# TUNING THE BURN-IN PHASE IN TRAINING RECURRENT NEURAL NETWORKS IMPROVES THEIR PERFORMANCE

**Julian D. Schiller**\*      **Malte Heinrich**      **Victor G. Lopez**      **Matthias A. Müller**
Leibniz University Hannover, Institute of Automatic Control, Hannover, Germany

## ABSTRACT

Training recurrent neural networks (RNNs) with standard backpropagation through time (BPTT) can be challenging, especially in the presence of long input sequences. A practical alternative to reduce computational and memory overhead is to perform BPTT repeatedly over shorter segments of the training data set, corresponding to truncated BPTT. In this paper, we examine the training of RNNs when using such a truncated learning approach for time series tasks. Specifically, we establish theoretical bounds on the accuracy and performance loss when optimizing over subsequences instead of the full data sequence. This reveals that the burn-in phase of the RNN is an important tuning knob in its training, with significant impact on the performance guarantees. We validate our theoretical results through experiments on standard benchmarks from the fields of system identification and time series forecasting. In all experiments, we observe a strong influence of the burn-in phase on the training process, and proper tuning can lead to a reduction of the prediction error on the training and test data of more than 60% in some cases.

## 1 INTRODUCTION

Recurrent neural networks (RNNs) are inherently suited for processing sequential data (Hewamalage et al., 2021; Lipton et al., 2015). For this reason, they have a wide range of applications, including natural language modeling (Khan et al., 2023; Mikolov et al., 2011; 2013), text and image generation (Sutskever et al., 2011; Gregor et al., 2015), and computer vision (Buhrmester et al., 2021), as well as classical time series tasks such as modeling dynamical systems (Pillonetto et al., 2025; Beintema et al., 2023; Forgione & Piga, 2021a;b; Schussler et al., 2019; Wang & Chen, 2006) and time series forecasting (Hewamalage et al., 2021; Elsworth & Güttel, 2020; Bianchi et al., 2017). While the Elman network (Elman, 1990) represents one of the most classic RNN architectures, prevailing approaches in practice combine the vanilla RNN with additional gating mechanisms, such as Long Short-Term Memory (LSTM) (Hochreiter & Schmidhuber, 1997) and Gated Recurrent Units (GRU) (Cho et al., 2014). RNNs have recently become a very active field of research again, driven also by the impressive performance that transformers have recently achieved for long-range reasoning (Kim et al., 2025). Modern architectures include deep state-space models (SSMs) such as S4 (Gu et al., 2022) and Mamba (Gu & Dao, 2024; Wang et al., 2025), as well as linear recurrent units (LRUs) (Orvieto et al., 2023; Zucchet et al., 2023). Other recent works involve Koopman theory (Li et al., 2025), sparse deep learning (Zhang et al., 2023), and probabilistic methods (Coscia et al., 2025).

RNNs are traditionally trained using backpropagation through time (BPTT) (Werbos, 1990); namely, in an iterative fashion using steepest descent. This requires to perform, at each iteration, a forward pass to compute the loss and a backward pass to compute the gradients, each running over the whole training sequence. However, in case the training sequence is large, many numerical problems can arise: the loss function landscape becomes increasingly complex (Allen-Zhu et al., 2019; Ribeiro et al., 2020), the computations of the forward and backward pass become computationally intense, and they may even blow up due to vanishing or exploding gradients (Bengio et al., 1994; Pascanu et al., 2013; Zucchet & Orvieto, 2024). A standard approach to ensure numerical stability, efficiency, and faster training due to less computational and memory overhead is to simply truncate the forward and backward passes, realizing truncated backpropagation through time (TBPTT) (Williams & Peng,

---

\*Corresponding author (e-mail: schiller@irt.uni-hannover.de); code is publicly available in the GitHub repository at https://github.com/jdschille/rnn-training

1990). This can be leveraged in a modified, stochastic version of the parameter update law (i.e., using stochastic gradient descent, SGD), where stochasticity comes from the fact that the loss gradients are computed over (a batch of) random segments of the training data set. Overall, TBPTT can be regarded as a practical compromise between learning effectiveness and computational effort. It is a standard method for RNN training, particularly in the presence of long training sequences (Mikolov et al., 2011; Tang & Glass, 2018; Aicher et al., 2020; Xu et al., 2023).

Computing the forward pass in TBPTT requires initializing the hidden state, which is usually set to zero or chosen at random. This may hinder the learning of long-term dependencies. Alternatives include propagating the hidden state across iterations (Xu et al., 2023; Elsworth & Güttel, 2020; Katrompas & Metsis, 2021) or estimating it using additional trainable autoencoders (Mohajerin & Waslander, 2017; Masti & Bemporad, 2021; Forgione et al., 2022; Beintema et al., 2021; 2023). While beneficial when long-term dependencies are crucial, zero initialization prevails in applications because the implementation is much simpler and renders the overall training process more efficient, numerically stable, and robust with respect to local minima (Xu et al., 2023; Allen-Zhu et al., 2019).

In this work, we consider the training of RNNs for time series tasks, relying on TBPTT with zero initialization to ensure a simple implementation and efficient training. To cope with internal transients resulting from initialization, we introduce the burn-in phase of the network as an additional parameter of the training process, effectively excluding them from the loss function. This was also previously suggested by Jaeger (2002); Bonassi et al. (2022); Beintema et al. (2021), referring to it (inconsistently) as "wash out" or "burn time"; however, without analyzing its influence on the resulting performance or providing systematic tuning guidelines, leaving it a rather unnoticed heuristic.

In this paper, we provide answers to the following questions.

1. *How much do we lose in the training by employing TBPTT with zero initialization?*
2. *How can we suitably modify the training process to reduce this loss?*

We establish novel theoretical results on the performance of RNN training when optimizing over subsequences of the training data using a truncated learning approach. More specifically, we estimate the performance gap (*regret*) with respect to a challenging benchmark RNN, trained by optimizing its hidden states on all of the training data. Regret is a standard measure for assessing algorithms in reinforcement learning (Jaksch et al., 2010; Agrawal et al., 2021), online optimal control (Agarwal et al., 2019; Li et al., 2019; Nonhoff & Müller, 2022; Martin et al., 2025), and state estimation (Brouillon et al., 2023; Schiller et al., 2025). Our technical analysis is based on viewing the training problem through the lens of optimal control, focusing in particular on the *turnpike phenomenon*. This property generally implies that optimal trajectories most of the time stay close to an optimal time-varying reference, which is regarded as the *turnpike*, see, e.g., Trélat & Zuazua (2025); Faulwasser & Grüne (2022); Zaslavski (2006); Carlson et al. (1991) for further details on this topic. Turnpike-related arguments have generally proven useful for analyzing the performance of optimization-based control and estimation techniques (Faulwasser et al., 2018; Grüne & Pirkelmann, 2019; Schiller et al., 2025).

The derived regret estimates depend on the hyperparameters of TBPTT, particularly the interplay between the internal forgetting rate of the network and the burn-in phase. These novel insights can be leveraged to select a suitable burn-in phase during training, ensuring that the resulting performance is close to that of the benchmark RNN. We validate our theoretical results in experiments using real world datasets from the fields of system identification and time series forecasting. In all experiments, we observe a strong influence of the burn-in phase on the training process, and proper tuning can lead to a reduction of the prediction error on the training *and test* data of more than 60% in some cases. The main findings of this paper for RNN training using TBPTT can be summarized as follows:

- *Theory:* We develop the concept of the burn-in phase from a heuristic to a theoretically grounded method.
- *Practice:* Tuning the burn-in phase can yield significant performance increase.

## 2 RECURRENT NEURAL NETWORKS

The core of any recurrent network is the internal dynamics of its hidden states. Given an input sequence $\{x_t\}_{t=1}^T$ of a certain length $T > 0$, the evolution of the hidden state $h_t$ for some pre-

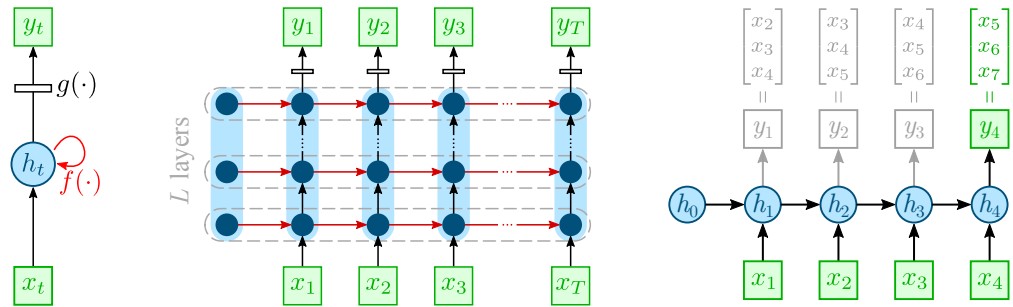

Figure 1: General structure of an RNN (left) with its internal architecture unfolded in space and time (middle). The network consists of $L$ recurrent layers and processes an input sequence of length $T$. Black arrows denote instantaneous spatial processing, red arrows denote temporal processing. Each arrow represents learnable linear or nonlinear projections and/or layers, potentially with additional skip connections, which can generally be captured by the functions $f$ and $g$. The right figure shows an exemplary RNN for time series forecasting applications, producing a forecast of $F = 3$ future values of a sequence $\{x_t\}$ using a lookback window of $N = 4$.

specified initial state $h_0$ can generally be described by the equation

$$h_t = f(h_{t-1}, x_t; \theta_{\mathrm{h}}), \quad t = 1, ..., T, \tag{1}$$

where $f$ is a function parameterized by the network parameters $\theta_{\mathrm{h}} \in \mathbb{R}^{n_h}$. The output $y_t$ of the network at time $t$ is produced through an output layer, which can be characterized in terms of a function $g$ such that

$$y_t = g(h_t, x_t; \theta_{\mathrm{y}}), \quad t = 1, ..., T, \tag{2}$$

where $\theta_{\mathrm{y}} \in \mathbb{R}^{n_y}$ corresponds to the parameters of the output layer. For convenience, throughout the following we consider the combined parameter vector $\theta = \begin{bmatrix} \theta_{\mathrm{h}}^\top, \theta_{\mathrm{y}}^\top \end{bmatrix}^\top \in \mathbb{R}^n$ with $n = n_h + n_y$, which consists of all the learnable parameters of the model in (1) and (2). Here, we want to emphasize that we consider a general multivariate setup, where $x_t \in \mathbb{R}^{d_x}$, $h_t \in \mathbb{R}^{d_h}$, and $y_t \in \mathbb{R}^{d_y}$.

Overall, the output $y_t$ at time $t = 1, ..., T$ depends on the initial hidden state $h_0$, the input sequence $\{x_j\}_{j=1}^t$, and the model equations (1)–(2) for a particular network parameter $\theta$. To make this explicit, we sometimes express the output $y_t$ in terms of the mapping $y_t(h_0, \theta, \{x_j\}_{j=1}^t)$ or, for convenience, using the full input sequence $y_t(h_0, \theta, \{x_j\}_{j=1}^T)$ (which is equivalent, as (1) ensures causality).

Note that the model in (1)–(2) constitutes a very general RNN description, which includes all commonly used network architectures, such as classical Elman RNNs (Elman, 1990), LSTMs (Hochreiter & Schmidhuber, 1997), GRUs (Cho et al., 2014), and echo state networks (Jaeger, 2002), but also recently proposed architectures such as LRU networks (Orvieto et al., 2023). In deep RNN structures, the hidden state $h_t$ corresponds to the stacked hidden states of all individual layers, and the function $f$ embodies the dynamics of the individual layers together with possibly additional nonlinear projections, gated linear units, and/or normalization operators between the layers, see also Figure 1. For the most simple case of a one-layer Elman RNN, the functions $f$ and $g$ and the parameter vector $\theta$ specialize to

$$f(h, x; \theta) = \phi(W_{\mathrm{hh}}h + W_{\mathrm{xh}}x + b_{\mathrm{h}}), \qquad g(h, x; \theta) = W_{\mathrm{hy}}h_t + b_{\mathrm{y}},$$
$$\theta = \begin{bmatrix} \mathrm{vec}(W_{\mathrm{hh}})^\top & \mathrm{vec}(W_{\mathrm{xh}})^\top & b_{\mathrm{h}}^\top & \mathrm{vec}(W_{\mathrm{hy}})^\top & b_{\mathrm{y}}^\top \end{bmatrix}^\top. \tag{3}$$

Here, the network dynamics (1) involve the weight matrix $W_{\mathrm{hh}}$ for the recurrent connection, the weight matrix $W_{\mathrm{xh}}$ between the input and the hidden layer, the bias vector $b_{\mathrm{h}}$, and the activation function $\phi$ (e.g., $\tanh$ or ReLU); the output (2) is formed by a linear layer with weight matrix $W_{\mathrm{hy}}$ and bias vector $b_{\mathrm{y}}$. All the weight matrices and bias vectors can be collected in the parameter vector $\theta$.

Note that we define the network output in (2) for all $t = 1, ..., T$. While this is natural in the context of dynamical systems, in many standard RNN applications such as classification tasks and time series forecasting, the output is usually only formed after the entire input sequence has been processed, that is, at time $T$. However, $g$ can still be evaluated at earlier times $t = 1, ..., T - 1$, see Figure 1. Finally, note that the RNN (1)–(2) covers any parameterized nonlinear state-space model; the subsequent analysis therefore directly also covers classical system identification problems (Ljung, 1999).

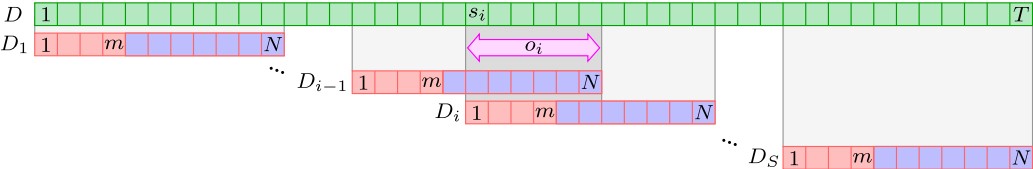

Figure 2: Data segmentation in TPBTT learning. Depicted are the full data sequence $D$ of length $T$ (green) and four (out of $S$) subsequences of length $N$ (red). The subsequences $D_i$ and $D_{i-1}$ start at samples $s_i$ and $s_{i-1}$, respectively, and result in the overlap $o_i$, compare (5). Blue highlighted segments appear in the loss function $L(\theta; D_i)$ in (7), which is determined by the burn-in phase length $m$.

# 3 TRAINING RNNS USING TBPTT AND MINI-BATCH SGD

In this section, we revisit the overall TBPTT learning algorithm to properly introduce our setup and notation. In particular, we illustrate the data segmentation process inherent to TBPTT (Section 3.1), define the segment-wise loss function (Section 3.2), show a basic mini-batch training scheme (Section 3.3), and present a suitable measure of training performance (Section 3.4). In this work, we consider the standard mean squared error (MSE) loss, where we treat the burn-in phase of the RNN as an additional hyperparameter of the training process (see Section 3.2). In Section 4, we establish novel regret guarantees for truncated learning, which exhibit a strong dependence on the burn-in phase.

## 3.1 DATA SEGMENTATION IN TBPTT

Let an input-output time series pair $D = \{(x_t^{\mathrm{d}}, y_t^{\mathrm{d}})\}_{t=1}^T$ of length $T$ be given and denote the input data sequence as $X^{\mathrm{d}} := \{x_t^{\mathrm{d}}\}_{t=1}^T$. TBPTT essentially implements the BPTT algorithm on short segments of the training data set. To this end, we divide the training sequence $D$ into $S$ subsequences, each of which has length $N < T$. We define the index sets for the subsequences as $\mathcal{S} := \{1, ..., S\}$ and for the elements within a subsequence as $\mathcal{N} := \{1, ..., N\}$. Moreover, we denote the input-output data points of subsequence $i \in \mathcal{S}$ that starts at data sample $s_i \in [1, T - N + 1]$ by $x_{j|i}^{\mathrm{d}} := x_{s_i+j-1}^{\mathrm{d}}$ and $y_{j|i}^{\mathrm{d}} := y_{s_i+j-1}^{\mathrm{d}}$ for all $j \in \mathcal{N}$. For convenience, these are grouped as follows:

$$D_i := \{(x_{j|i}^{\mathrm{d}}, y_{j|i}^{\mathrm{d}})\}_{j=1}^N, \quad X_i^{\mathrm{d}} := \{x_{j|i}^{\mathrm{d}}\}_{j=1}^N, \quad i \in \mathcal{S}. \tag{4}$$

In order to make use of all available data during training, in the following we restrict ourselves to the case where the subsequences $D_i$ cover the complete data set $D$, and furthermore, assume that the start samples $s_i$, $i \in \mathcal{S}$ are arranged in ascending order. Hence, a subsequence $D_i$ may share some data points with the preceding subsequence $D_{i-1}$, which can be characterized by the overlap $o_i$ as

$$o_i := N - (s_i - s_{i-1}), \quad i = 2, ..., S. \tag{5}$$

Here, we note that $o_i$ takes values in the interval $[0, N - 1]$. Considering all $S$ subsequences, we define the minimum overlap according to

$$o_{\min} := \min_{2 \leq i \leq S} o_i = N - \max_{2 \leq i \leq S} (s_i - s_{i-1}). \tag{6}$$

The overall data segmentation process is also illustrated in Figure 2. The variables $S$, $N$, and $s_i$, $i \in \mathcal{S}$, represent important hyperparameters of the TBPTT algorithm that have a strong influence on the training process and the resulting performance of the trained RNN. In general, suitable values may be chosen by leveraging additional information about the given data set and the underlying application (such as specific patterns or seasonal components of different length). If no such information is available, for a fixed value of $N$, a practical choice is to select $S = T - N + 1$ (i.e., use as many subsequences of length $N$ as possible), which implies that $s_i = i$ for all $i \in \mathcal{S}$ and yields the maximum overlap $o_i = N - 1 = o_{\min}$ uniformly for all $2 \leq i \leq S$.

## 3.2 SEGMENT-WISE LOSS FUNCTION WITH BURN-IN PHASE

In this paper, we consider the MSE loss function—a standard choice for time series tasks. Extensions to other loss functions suitable for, e.g., classification problems are an interesting topic of future work.

For a particular parameter $\theta$, we define the loss function involving the data subsequence $D_i$ as

$$L(\theta; D_i) = \frac{1}{N - m} \sum_{j=m+1}^{N} \|y_j(0, \theta, X_i^{\mathrm{d}}) - y_{j|i}^{\mathrm{d}}\|^2, \quad i \in \mathcal{S}. \tag{7}$$

The variable $m$ appearing in the lower bound of summation in (7) refers to the *burn-in phase* of the network, see Figure 2. It is an additional design variable to compensate for network outputs that might be negatively affected by internal transients resulting from the zero initialization. It can be chosen from the interval $[0, N - 1]$, where $m = 0$ recovers the standard MSE loss (without burn-in phase). Similar approaches have been used previously (Jaeger, 2002; Bonassi et al., 2022; Beintema et al., 2021), albeit without performing any detailed analysis or providing tuning guidelines. Note that $m = N - 1$ captures scenarios in which the output is generated by default only after processing the entire input subsequence $X_i$ of length $N$, as is the case with time series forecasting, compare Figure 1 (right). Nevertheless, in the following sections we show both theoretically and experimentally that even here, a choice of $0 < m < N - 1$ can be beneficial for overall performance, suggesting that $m$ should generally be considered a hyperparameter of RNN training.

## 3.3 BASIC TRAINING ALGORITHM

The network parameter $\theta$ are usually trained iteratively with SGD using mini-batches (Lipton et al., 2015; Hewamalage et al., 2021; Tian et al., 2023). More specifically, at each iteration $k$, we choose a random subset $\mathcal{B}_k \subset \mathcal{S}$ of size $|\mathcal{B}_k| =: b < S$ to update $\theta$ according to

$$\theta_{k+1} = \theta_k - \eta_k \Delta\theta_k, \qquad \Delta\theta_k := \frac{1}{b} \sum_{i \in \mathcal{B}_k} \nabla_\theta L(\theta_k; D_i). \tag{8}$$

Here, $\Delta\theta_k$ represents the batch-averaged gradient. The learning rate $\eta_k > 0$ is usually chosen to be a small constant or adapted with time (Tian et al., 2023). The learning procedure is summarized in Algorithm 1. Here, Step 4 actually corresponds to an additional loop that performs, for each subsequence $i$ contained in the current mini-batch $\mathcal{B}_k$, a forward pass to compute the segment-wise loss $L(\theta_k; D_i)$, and a backward pass to compute the corresponding gradient. However, these computations are completely independent for each $i \in \mathcal{B}_k$ and can therefore be performed in parallel (possibly on a GPU), which significantly accelerates the training. The basic algorithm can be adapted, e.g., by employing SGD variants such as AdaGrad, RMSProp, Adam, etc. (Tian et al., 2023; Bottou & Bousquet, 2007; Kingma & Ba, 2015), all of which are part of standard deep learning libraries such as PyTorch. Recent results on the global convergence and stability of SGD algorithms in the context of machine learning can be found in the work of Patel et al. (2022); Allen-Zhu et al. (2019).

---

**Algorithm 1** RNN training using TBPTT with mini-batch SGD

---
**Require:** data set $D$, learning rate $\eta_k > 0$
1: **Initialize** $\theta_0 \in \mathbb{R}^n$, $k = 0$.
2: **while** algorithm not converged **do**
3:      Sample $\mathcal{B}_k \subset \mathcal{S}$                                      ▷ Select current mini-batch
4:      $\Delta\theta_k = \frac{1}{b} \sum_{i \in \mathcal{B}_k} \nabla_\theta L(\theta_k; D_i)$                  ▷ Compute batch-averaged gradient
5:      $\theta_{k+1} = \theta_k - \eta_k \Delta\theta_k$                               ▷ Update parameter
6:      $k = k + 1$
7: **end while**

---

**Remark 1.** *Algorithm 1 employs zero initialization of each segment-wise forward pass, which is the most classical form of TBPTT. A slight modification is to preserve the chronological order of the subsequences when selecting the current batch $\mathcal{B}_k$ in Step 3 and pass the hidden states from iteration $k$ to $k + 1$, which is sometimes referred to as "stateful" training (Katrompas & Metsis, 2021; Elsworth & Güttel, 2020; Xu et al., 2023). Although this can be advantageous when long-term dependencies are crucial, the former approach prevails in deep learning applications because the randomness of the training algorithm can help escape from local minima and generally renders the training process more numerically stable, robust, and efficient (Xu et al., 2023; Allen-Zhu et al., 2019). Still, extending our theory to stateful training could be an interesting topic for future work.*

### 3.4 TRAINING PERFORMANCE

Recall that TBPTT is a practical approximation of BPTT using the full training data set $D$. Hence, it seems meaningful to measure the performance of the training algorithm by re-evaluating the network after successful training, processing the full training input sequence $X^{\mathrm{d}}$ and computing the MSE between the generated outputs and the ground truth data $Y^{\mathrm{d}}$. Specifically, for a given $\theta$ and initial hidden state $h_0$, we define

$$P(h_0, \theta; D) := \frac{1}{T - m} \sum_{t=m+1}^{T} \|y_t(h_0, \theta, X^{\mathrm{d}}) - y_t^{\mathrm{d}}\|^2. \tag{9}$$

To ensure a fair comparison between parameter sets when using different network initializations, we in fact consider the truncated MSE in (9), where, similar to the segment-wise loss in (7), we exclude internal network transients by incorporating the burn-in phase $m$.

## 4 REGRET GUARANTEES FOR TRUNCATED LEARNING

This section contains our theoretical results, providing answers to the research questions stated in Section 1. Our results indicate how the design parameters $S$ (number of the subsequences), $N$ (length of the subsequences), and burn-in phase length $m$ of the network influence the training performance of RNNs when pursuing a truncated learning approach. Here, we assess the training in terms of the achieved performance gap (i.e., the *regret*) with respect to a challenging benchmark.

### 4.1 AN OPTIMIZATION PERSPECTIVE

The averaged gradient $\Delta\theta_k$ in (8) can be interpreted as an unbiased estimate of the gradient of the full-batch loss function, that is, the right-hand side of (8) with $\mathcal{B}$ and $b$ replaced by $\mathcal{S}$ and $S$, respectively, compare Tian et al. (2023). Hence, TBPTT as implemented by Algorithm 1 actually aims at solving the following optimization problem:

$$V^* = \min_{\theta} \frac{1}{S} \sum_{i=1}^{S} L(\theta, D_i) = \min_{\theta} \frac{1}{S(N - m)} \sum_{i=1}^{S} \sum_{j=m+1}^{N} \|y_j(0, \theta, X_i^{\mathrm{d}}) - y_{j|i}^{\mathrm{d}}\|^2. \tag{10}$$

The optimization objective in (10) involves $S$ hidden state sequences of length $N + 1$, each initialized with zero, coupled by the same parameterization $\theta$. This is conceptually different from the definition of the performance metric in (9), which assesses the training performance based on a *single hidden state sequence of length* $T + 1$. This naturally raises the question: What if we leverage this additional knowledge during optimization? This motivates considering the following benchmark problem:

$$V^{\mathrm{b}} = \min_{\{h_t\}_{t=0}^{T}, \theta} \frac{1}{S(N - m)} \sum_{i=1}^{S} \sum_{j=m+1}^{N} \|y_j(h_{s_i}, \theta, X_i^{\mathrm{d}}) - y_{j|i}^{\mathrm{d}}\|^2 \tag{11a}$$

$$\text{subject to} \quad h_t = f(h_{t-1}, x_t^{\mathrm{d}}; \theta), \quad t = 1, ..., T. \tag{11b}$$

Note that the benchmark problem in (11b) uses the original TBPTT optimization objective from (10), i.e., aims at minimizing the averaged MSE of $S$ output sequences of length $N$. However, the additional constraint in (11b) ensures that the underlying hidden state sequences actually correspond to different subsequences of a *single* long hidden state sequence $\{h_t\}_{t=0}^{T}$. Comparing the solutions of (10) and (11) therefore allows us to investigate and quantify the effects of using zero initialization in TBPTT instead of the optimal initializations. Since both problems share the same optimization objective, we can perform this comparison in terms of: 1) the *training regret*, quantifying the optimality gap $V^* - V^{\mathrm{b}}$ (see Theorem 1); and 2) the *performance regret*, where we apply the parameterized networks to the full training input sequence and evaluate the performance metric (9) (see Theorem 2).

In the following, we refer to the solution of the TBPTT problem (10) with the parameter $\theta^*$, and to the benchmark problem in (11) with the hidden state sequence $\{h_t^{\mathrm{b}}\}_{t=0}^{T}$ and network parameter $\theta^{\mathrm{b}}$.

### 4.2 MAIN RESULTS

In this section, we present novel accuracy and regret guarantees for truncated learning. To provide insights into the training process and its tuning knobs that are also valuable to practitioners, we focus

on the key messages and defer most technical details to the appendix; moreover, we formulate our theorems rather informally in the main body of the paper, while they are made precise in Appendix A.

In line with standard learning techniques, we suppose that the training data is normalized in the sense that $(x_t^{\mathrm{d}}, y_t^{\mathrm{d}}) \in \mathcal{X}^{\mathrm{d}} \times \mathcal{Y}^{\mathrm{d}}$ for all $t = 1, ..., T$, where the sets $\mathcal{X}^{\mathrm{d}} \subset \mathbb{R}^{d_x}$ and $\mathcal{Y}^{\mathrm{d}} \subset \mathbb{R}^{d_y}$ are compact. Usual design choices include the intervals $[0, 1]$ or $[-1, 1]$ for the elements of each data point. We focus on parameterizations that result in the RNN having the following output stability property.

**Assumption 1.** *There exists a (non-empty) set $\Theta \subset \mathbb{R}^n$ such that, for all $\theta \in \Theta$, the network (1)–(2) is exponentially incrementally output stable along input sequences of the training data, that is, there exist constants $C > 0$ and $\lambda \in (0, 1)$ such that*

$$\|y_t(h_0^{(1)}, \theta, X) - y_t(h_0^{(2)}, \theta, X)\| \leq C\lambda^t \|h_0^{(1)} - h_0^{(2)}\|, \; t = 1, 2, ..., T' \tag{12}$$

*for any two initial hidden states $h_0^{(1)}, h_0^{(2)} \in \mathbb{R}^n$ and any sequence $X$ of length $T' \in [0, T]$ extracted from the training input sequence $X^{\mathrm{d}}$.*

Assumption 1 essentially requires that the RNN "forgets" its initialization over time, so that for the same input sequence $X$, it always produces output sequences that converge to each other, despite different initial hidden states. This is a standard condition (Ljung, 1978), and moreover, generally represents a desirable feature in the context of RNNs. Indeed, when applying an RNN in practice, it should always generate meaningful outputs that are not overly sensitive to its initialization. We want to emphasize that Assumption 1 does *not* refer to a property of some underlying (unknown) data-generating system—it is rather a property of the trained RNN model. Consequently, one could directly enforce it during training by normalizing and clipping certain RNN weights when performing the gradient updates such that the RNN forward dynamics essentially become stable (i.e., contractive), see Miller & Hardt (2019) for more details and corresponding methods (applicable to, e.g., LSTMs) and compare also Bonassi et al. (2021); Kolter & Manek (2019); Kozachkov et al. (2022). Nevertheless, invoking these sufficient conditions is generally conservative, potentially limiting expressiveness of the RNN. However, note also that contractive dynamics are not necessary for Assumption 1—rather, it suffices that the trained model satisfies the property in (12) only along input sequences of the training data $X^{\mathrm{d}}$, which can be easily tested numerically after training, compare Section 5.

Our first result bounds the gap between solutions to the TBPTT problem (10) and the benchmark (11).

**Theorem 1** (informal). *Let the TBPTT solution $\theta^*$ of the problem in (10) and the benchmark solution $\theta^{\mathrm{b}}$ of the problem in (11) be given and assume that both parameters render the respective RNN incrementally output stable in the sense of Assumption 1. Under an additional technical condition, there exist constants $C_1, C_2 > 0$ such that the following properties hold:*

1. *Output accuracy:* $\displaystyle \sum_{j=m+1}^{N} \frac{1}{S} \sum_{i=1}^{S} \|y_j(0, \theta^*, X_i^{\mathrm{d}}) - y_j(h_{s_i}^{\mathrm{b}}, \theta^{\mathrm{b}}, X_i^{\mathrm{d}})\|^2 \leq C_1 \lambda^m$

2. *Training regret:* $\displaystyle V^* - V^{\mathrm{b}} \leq C_2 \cdot \frac{\lambda^m}{N - m}$

*Here, $\lambda \in (0, 1)$ is from Assumption 1.*

We provide the proof of Theorem 1 in Appendix A.3 (Corollary 1). The first property in Theorem 1 states that the sum of the averaged squared errors between the predictions generated by the TBPTT solution $\theta^*$ and the benchmark solution $\theta^{\mathrm{b}}$ along all subsequences $S$ is bounded by a constant. As this constant does *not* depend on $N$, this sum must become smaller if $N$ is increased, which necessarily implies that the TBPTT outputs get closer to the benchmark outputs. Moreover, the bound gets smaller for larger values of $m$ (as the outputs are less influenced by transients) and for smaller values of $\lambda$ (as the outputs converge faster). The second property given in Theorem 1 reveals that the optimal value for $m$ when aiming for a small regret bound is in fact highly dependent on the convergence rate $\lambda$; for slow convergence (larger values of $\lambda$), smaller values of $m$ are preferable, and *vice versa*.

**Remark 2.** *The statement of Theorem 1 (and Theorem 2 below) involves "an additional technical condition". As we show in detail in Appendix A, we need to exclude cases where the optimal output sequences fulfill a certain geometric relation, leading to the same value functions despite being different. However, this is rather of formal interest, since 1) it can be avoided in advance by choosing a suitable RNN architecture; 2) we conjecture that our theory can be generalized to these cases as well while preserving the key implications. For further details, see Remark 4 in Appendix A.2.*

Our second result assesses the trained RNN when processing the full training input sequence $X^{\mathrm{d}}$.

**Theorem 2** (informal). *Let the TBPTT solution $\theta^*$ of the problem in (10) and the benchmark solution $\theta^{\mathrm{b}}$ of the problem in (11) be given and assume that both parameters render the respective RNN incrementally output stable in the sense of Assumption 1. Let the burn-in phase parameter $m$ satisfy $m \leq o_{\min}$, where $o_{\min}$ is the minimum overlap between subsequences as defined in (6). Under an additional technical condition, there exist some $E_1, E_2 > 0$ such that the following properties hold:*

1. *Output accuracy:* $\quad \dfrac{1}{T-m} \displaystyle\sum_{t=m+1}^{T} \|y_t(0, \theta^*, X^{\mathrm{d}}) - y_t(h_0^{\mathrm{b}}, \theta^{\mathrm{b}}, X^{\mathrm{d}})\|^2 \leq E_1 \dfrac{(S-1)\lambda^{2o_{\min}} + S\lambda^m}{T-m}$

2. *Performance regret:* $\quad P(0, \theta^*; D) - P(h_0^{\mathrm{b}}, \theta^{\mathrm{b}}; D) \leq E_2 \cdot \sqrt{\dfrac{(S-1)\lambda^{2o_{\min}} + S\lambda^m}{T-m}}$

*Here, $\lambda \in (0,1)$ is from Assumption 1 and $P$ corresponds to the training performance defined in (9).*

Theorem 2 is proven in Appendix A.4 (Theorem 3). It reveals similar dependencies of the tuning parameters on the training performance as Theorem 1, ensuring that the output sequence generated by the TBPTT solution is close to that of the benchmark, also when processing the full input sequence. However, there are also subtle differences: First, there is an additional condition requiring $m \leq o_{\min}$. This ensures that the full-batch loss functions used in (10) and (11) involve all data samples $\{y_t^{\mathrm{d}}\}_{t=m+1}^T$. Second, there is an additional influence of the number of training subsequences $S$. Still, the regret bounds are mainly influenced by the parameters $m$ and $\lambda$, whereby smaller values of $m$ are preferable for $\lambda \to 1$ and *vice versa*. If an upper bound for $\lambda$ is enforced during training, the optimal value of $m$ minimizing the regret estimates from Theorems 1 and 2 could, in principle, be computed explicitly. However, the estimates are generally conservative, as Assumption 1 (on which they are based) also covers the slowest possible decay and is therefore inherently conservative in *quantitative* terms. Our theoretical results should therefore be understood as *qualitative* tuning guidelines.

## 5 EXPERIMENTS

We validate our theory on various time series tasks (the code is available online at https://github.com/jdschille/rnn-training). We first consider a small-scale problem of academic nature in Section 5.1, where we rely on synthetic data. This allows us to solve the benchmark problem (11) using direct nonlinear optimization methods and hence to illustrate Theorems 1 and 2. In Section 5.2, we show that the obtained insights carry over to RNN training applied to public data sets from the fields of system identification and time series forecasting, using classical TBPTT and mini-batch SGD methods. Here, we numerically tested the TBPTT-trained models for compliance with Assumption 1, which was always fulfilled with $\lambda = 0.99, \ldots, 0.9999 < 1$. However, as the benchmark problem (11) can no longer be directly solved due to the size of the problem, we assess RNN performance by considering the MSE of the generated outputs with respect to training and test data and additionally compare it with BPTT and stateful (see Remark 1) TBPTT training results.

### 5.1 SYNTHETIC DATA

We aim to model an unknown linear single-input single-output system, for which a noisy training data set of length $T = 100$ is available (see Figure 5 in Appendix B.1). To this end, we design a linear RNN (Proca et al., 2025) with a one-dimensional hidden state, corresponding to the model from (3) with $\phi(x) = x$ and $b_{\mathrm{h}} = b_{\mathrm{y}} = 0$, where we additionally enforce $\|W_{\mathrm{hh}}\| \leq 0.999$ during optimization to ensure Assumption 1. For training, we select $N = 21$, set $S = T - N + 1 = 80$ (compare Section 3.1), and compute the solutions to the TBPTT problem (10) and benchmark problem (11) for different values of $m$ using the nonlinear programming solver IPOPT (Wächter & Biegler, 2005) in MATLAB. The key training results depicted in Figure 3 are in line with our theoretical results provided in Section 4.2; specifically, the left plot shows that the $j$-step batch-averaged output error $e_j := \frac{1}{S} \sum_{i=1}^{S} \|y_j(0, \theta^*, X_i^{\mathrm{d}}) - y_j(h_{s_i}^{\mathrm{b}}, \theta^{\mathrm{b}}, X_i^{\mathrm{d}})\|^2$ converges to a neighborhood of zero, the size of which decreases with increasing $m$. Hence, larger values of $m$ allow for longer transient phases, and the RNN effectively exploits this during training to get closer to the benchmark outputs. As indicated by the right plot of Figure 3, this behavior is also reflected in the resulting performance achieved over the training and an additional test data set (compare Appendix B.1). Here, the regret exponentially converges to zero with increasing $m$, and almost zero regret can be achieved when using $m \geq 12$.

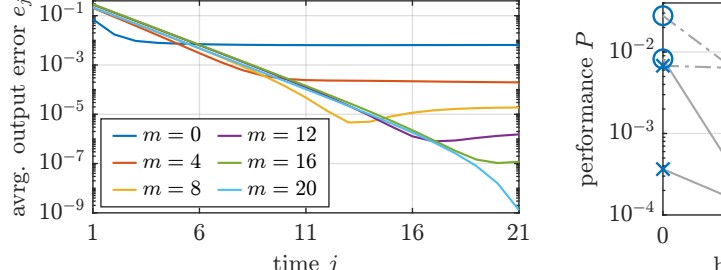

Figure 3: Training results for the synthetic data experiment for different values of the burn-in phase $m$. Left: batch-averaged output error $e_j$. Right: performance $P$ (the truncated MSE from (9)) achieved by solving the TBPTT problem (10) ('∘' markers) and the benchmark problem (11) ('×' markers).

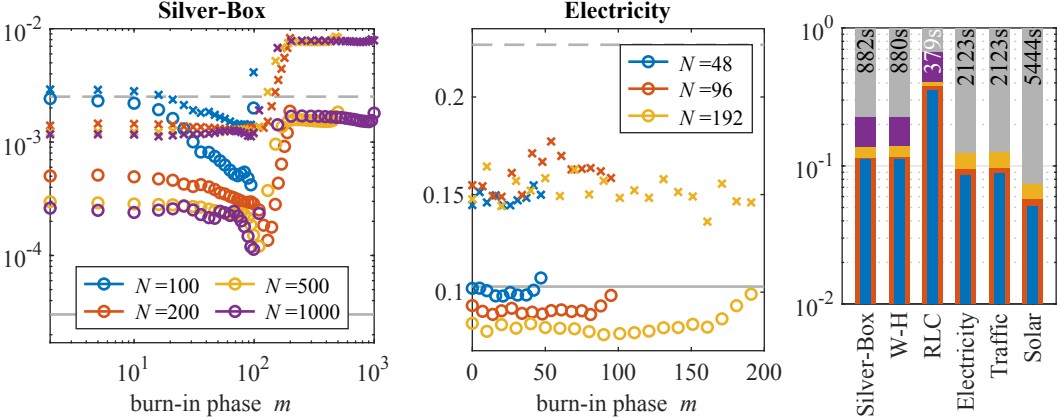

Figure 4: Training results. Left and middle: MSE of RNN predictions for training ('∘') and test data ('×') after (re-)training; gray lines correspond to the BPTT-trained RNN evaluated on the training (solid) and test data (dashed). Right: Averaged TBPTT training times relative to BPTT.

## 5.2 DATA SETS FROM SYSTEM IDENTIFICATION AND TIME SERIES FORECASTING

In this section, we consider the training of RNNs on public data sets from the fields of 1) system identification: Silver-Box (Wigren & Schoukens, 2013), Wiener-Hammerstein (W-H) (Schoukens et al., 2009), RLC circuit (RLC) (Forgione & Piga, 2021a); and 2) time series forecasting: Electricity, Traffic, Solar-Energy (Lai et al., 2018; Wang et al., 2025).

For system identification, we aim to design an RNN as a black-box model of the state-space representation of the respective underlying dynamical system. Here, we choose an LSTM cell with hidden state dimension $d_h = 8$ and a linear output layer, see Appendix B for further details. For each data set, we train multiple RNN instances using TBPTT and various combinations of window length $N$ and burn-in phase $m$. In Figure 4 (left) and Figure 6 in Appendix B, we observe a strong influence of $m$ on the MSE achieved on the training and test data. Interestingly, the results show very similar qualitative behavior when varying $m$, regardless of $N$. The training results are summarized in Table 1 (upper half), where we chose $m^*$ based on an additional validation data set and refer to $\bar{m} = 0$ as the baseline choice. Consequently, proper tuning of $m$ can yield a significant reduction of the MSE on both training and test data compared to baseline TBPTT, narrowing the gap to BPTT performance on the training data while avoiding the risk of overfitting and reducing the training time (see Figure 4).

For time series forecasting, we design an RNN to predict $F$ future values of the time series contained in the lookback window of length $N$, compare Figure 1 (right). Here, we choose $F = 96$ and consider lookback windows of different lengths $N = 48, 96, 192$, compare Wang et al. (2025). To simplify the training, we restrict ourselves to univariate forecasting and consider one LSTM cell with hidden state dimension $d_h = 96$ and a linear output layer. For each data set, we train multiple RNN instances for different values of $N$ and $m$. To this end, we split the available time series into training, validation,

Table 1: Training on public system identification (sysid) and time series forecasting (TSF) data sets.

| | $N$ | $m^*$ | MSE on training data | | | | MSE on test data | | | |
|---|---|---|---|---|---|---|---|---|---|---|
| | | | TBP $\bar{m}$ | TBP $m^*$ | TBP sf | BP | TBP $\bar{m}$ | TBP $m^*$ | TBP sf | BP |
| Silver-Box | 100 | 94 | 0.242 | 0.042 (-83%) | 0.059 | 0.003 | 0.291 | 0.142 (-51%) | 0.222 | 0.251 |
| | 200 | 94 | 0.050 | 0.030 (-41%) | 0.048 | 0.003 | 0.140 | 0.135 (-4%) | 0.221 | 0.251 |
| | 500 | 94 | 0.030 | 0.019 (-37%) | 0.051 | 0.003 | 0.121 | 0.122 (+1%) | 0.185 | 0.251 |
| | 1000 | 94 | 0.026 | 0.012 (-54%) | 0.032 | 0.003 | 0.117 | 0.112 (-4%) | 0.146 | 0.251 |
| W-H | 100 | 21 | 0.220 | 0.153 (-31%) | 0.541 | 0.063 | 0.511 | 0.488 (-4%) | 1.014 | 0.445 |
| | 200 | 36 | 0.168 | 0.138 (-18%) | 0.514 | 0.063 | 0.454 | 0.476 (+5%) | 0.961 | 0.445 |
| | 500 | 0 | 0.135 | 0.135 (±0%) | 0.465 | 0.063 | 0.397 | 0.397 (±0%) | 0.875 | 0.445 |
| | 1000 | 0 | 0.116 | 0.116 (±0%) | 0.391 | 0.063 | 0.372 | 0.372 (±0%) | 0.737 | 0.445 |
| RLC | 100 | 83 | 2.290 | 1.122 (-51%) | 1.101 | 0.107 | 3.527 | 1.093 (-69%) | 1.183 | 0.176 |
| | 200 | 110 | 0.971 | 0.309 (-68%) | 1.057 | 0.107 | 1.462 | 0.307 (-79%) | 1.186 | 0.176 |
| | 500 | 130 | 0.442 | 0.186 (-58%) | 0.716 | 0.107 | 0.538 | 0.228 (-58%) | 0.932 | 0.176 |
| | 1000 | 110 | 0.306 | 0.176 (-42%) | 0.674 | 0.107 | 0.402 | 0.362 (-10%) | 0.871 | 0.176 |
| Electr. | 48 | 10 | 0.107 | 0.101 (-6%) | 0.165 | 0.103 | 0.150 | 0.146 (-3%) | 0.418 | 0.227 |
| | 96 | 14 | 0.098 | 0.089 (-10%) | 0.160 | 0.103 | 0.159 | 0.149 (-6%) | 0.279 | 0.227 |
| | 192 | 20 | 0.099 | 0.083 (-16%) | 0.152 | 0.103 | 0.146 | 0.144 (-1%) | 0.219 | 0.227 |
| Traffic | 48 | 0 | 0.297 | 0.287 (-3%) | 0.321 | 0.240 | 0.343 | 0.316 (-8%) | 0.728 | 0.498 |
| | 96 | 75 | 0.228 | 0.234 (+3%) | 0.331 | 0.240 | 0.305 | 0.280 (-8%) | 0.813 | 0.498 |
| | 192 | 20 | 0.251 | 0.205 (-18%) | 0.370 | 0.240 | 0.284 | 0.276 (-3%) | 0.359 | 0.498 |
| Solar | 48 | 31 | 0.300 | 0.312 (+4%) | 0.312 | 0.152 | 0.250 | 0.257 (+3%) | 0.772 | 0.110 |
| | 96 | 75 | 0.068 | 0.069 (+2%) | 0.203 | 0.152 | 0.227 | 0.184 (-19%) | 0.349 | 0.110 |
| | 192 | 141 | 0.020 | 0.020 (-1%) | 0.185 | 0.152 | 0.211 | 0.112 (-47%) | 0.092 | 0.110 |

RNNs are evaluated over the interval $[1, T]$ (sysid) or lookback windows of length $N$ (TSF); "TBP $\bar{m}$": baseline TBPTT training using $m=\bar{m}=0$ (sysid) and $m=\bar{m}=N-1$ (TSF); "TBP $m^*$": TBPTT training with $m=m^*$, where values in parentheses denote relative change with respect to the respective baseline; "TBP sf": stateful TBPTT training using $m=\bar{m}$; "BP": full BPTT training using $m=0$. The MSE values corresponding to system identification datasets (Silver-Box, W-H, RLC) are scaled by a factor of $10^2$.

and test segments, where we first (pre-)train the RNN on the training segment, tune $m^*$ by additionally considering the validation segment, re-train the RNN using the combined training and validation segments, and ultimately evaluate the forecast performance on the test segment, see Appendix B.2 for further details. Figure 4 (middle) and Figure 6 in Appendix B.2 show the forecast MSE after re-training the RNN using the considered combinations of $N$ and $m$. Here, we recall that $m = N - 1$ corresponds to the standard configuration in time series forecasting (compare the discussion at the end of Section 3.2), which we consider as the the baseline TBPTT with corresponding value $\bar{m}$. In the bottom half of Table 1, we compare the MSE achieved after training using $m = m^*$ with the baseline TBPTT, stateful TBPTT (i.e., passing hidden states over lookback windows during training), and full BPTT using $m = 0$ (which corresponds to inferring the forecast $F$ using all available past data). Again, we find that tuning $m^*$ is beneficial overall, as we can achieve a significant reduction in the training and/or test MSE in the majority of cases. Choosing $m < \bar{m}$ can be interpreted as additional regularization, which turns out to be numerically advantageous when performing the gradient updates during training. The fact that TBPTT with zero initialization outperforms stateful TBPTT and BPTT training in some cases in Table 1 is due to the positive effect of randomness in the training process (the data batches are shuffled, see Remark 1), which renders the training more numerically stable and yields faster convergence (note that we use the same number of iterations for all training methods).

## 6 CONCLUSIONS

In this paper, we developed the concept of burn-in phase from a heuristic to a theoretically grounded method for RNN training. Our results indicate how the training hyperparameters—in particular, the length of the burn-in phase—influence the training performance. In experiments on public data sets, we showed that tuning the burn-in phase improves RNN performance compared to standard choices.

## ACKNOWLEDGMENTS

This work was supported by the Deutsche Forschungsgemeinschaft (DFG, German Research Foundation) as part of the Research Unit "Active Learning for Systems and Control (ALeSCo)" – project number 535860958.

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

## A  THEORETICAL ANALYSIS

This section contains proofs and further technical details of Theorems 1 and 2 stated in Section 4.2. Our analysis builds on a generalized optimization problem (Section A.1), which captures the TBPTT problem from (10) and the benchmark problem from (11). In Section A.2, we show that this generalized problem exhibits an output turnpike property, which is used in Sections A.3 and A.4 to prove Theorems 1 and 2, respectively.

### A.1  THE LIFTED OPTIMIZATION PROBLEM

To allow for a precise technical analysis, we re-formulate the problems (10) and (11) in a multiple shooting fashion (see, e.g., Section 8.5 of the book by Rawlings et al. (2020) for further details on this topic). More specifically, we implement the output mapping used in the respective loss functions as part of the problem constraints and introduce additional decision variables.

To this end, we define the containers of the predicted hidden state and output sequences as

$$H := \{h_{j|i}\}_{\forall j \in \mathcal{N}_0, \forall i \in \mathcal{S}}, \quad Y := \{y_{j|i}\}_{\forall j \in \mathcal{N}, \forall i \in \mathcal{S}}, \tag{13}$$

where $\mathcal{N}_0 := \mathcal{N} \cup \{0\}$. Furthermore, we define the stage cost $l_{j|i} : \mathbb{R}^{d_y} \to \mathbb{R}_{\geq 0}$ associated with stage $j \in \mathcal{N}$ and subsequence $i \in \mathcal{S}$ according to

$$l_{j|i}(y) = \|y - y_{j|i}^{\mathrm{d}}\|^2. \tag{14}$$

Now, we can express the loss functions used in (10) and (11) as

$$J_S(Y; D) := \frac{1}{S(N-m)} \sum_{i=1}^{S} \sum_{j=m+1}^{N} l_{j|i}(y_{j|i}) \tag{15}$$

and formulate the following optimization problem:

$$\min_{H,Y,\theta} J_S(Y; D) \tag{16a}$$

$$\text{subject to} \quad h_{j|i} = f(h_{j-1|i}, x_{j|i}^{\mathrm{d}}; \theta), \ \forall j \in \mathcal{N}, \ \forall i \in \mathcal{S}, \tag{16b}$$

$$y_{j|i} = g(h_{j|i}, x_{j|i}^{\mathrm{d}}; \theta), \ \forall j \in \mathcal{N}, \ \forall i \in \mathcal{S}, \tag{16c}$$

$$\{h_{j|i}\}_{j=0}^{N} \in \mathcal{H}^{N+1}, \ \{y_{j|i}\}_{j=1}^{N} \in \mathcal{Y}^{N}, \quad \forall i \in \mathcal{S} \tag{16d}$$

$$\theta \in \Theta, \tag{16e}$$

$$R(H) = 0. \tag{16f}$$

The constraints in (16b) and (16c) imply that over each subsequence $i \in \mathcal{S}$, the corresponding hidden state and output sequences satisfy the network equations from (1) and (2) for all $j \in \mathcal{N}$. Condition (16d) ensures that these sequences have elements from the sets $\mathcal{H} \subset \mathbb{R}^{d_h}$ and $\mathcal{Y} \subset \mathbb{R}^{d_y}$, respectively, where we assume that $\mathcal{H}$ and $\mathcal{Y}$ are compact. Note that this is required for technical reasons in the proof of our results below and in particular does not need to be incorporated in, e.g., Algorithm 1 (as suitable sets $\mathcal{H}$ and $\mathcal{Y}$ can be found in any case after successful training, compare also Remark 3 below). The constraint in (16e) ensures that the resulting parameter $\theta$ leads to a network model that is incrementally output stable in the sense of Assumption 1.

Depending on the last constraint in (16f), we consider three different settings:

1. The TBPTT case:

$$R(H) = [h_{0|1}, ..., h_{0|S}]. \tag{17}$$

This results in each initial hidden state $h_{0|i}$, $i \in \mathcal{S}$, being initialized to zero. From a theoretical perspective, the problem (16) under (17) is fully equivalent to the original TBPTT problem in (10) (provided that the solution to the latter satisfies Assumption 1, i.e., $\theta^* \in \Theta$). We denote the solution to (16) under (17) as $(H^*, Y^*, \theta^*)$.

2. The coupled case:

$$R(H) = [h_{0|2} - h_{N-o_2|1}, ..., h_{0|S} - h_{N-o_S|S-1}]. \tag{18}$$

This condition implies that for each $i \in \mathcal{S}$ with $i \geq 2$, the corresponding initial hidden state $h_{0|i}$ is equal to the hidden state $h_{N-o_i|i-1}$, where $o_i$ corresponds to the overlap between the subsequences $i$ and $i - 1$ as defined in (5). As a consequence, all hidden state sequences contained in $H$ are a subsequence of the same global hidden state sequence of length $T + 1$, where the initial hidden state state $h_{0|1}$ is left as a free optimization variable. We denote the solution to (16) under (18) as $(H^c, Y^c, \theta^c)$. Again, from a theoretical point of view, the problem formulation in (16) under (18) is fully equivalent to the benchmark problem in (11) (provided that $\theta^b \in \Theta$). In particular, their solutions sets coincide in the sense that $\theta^b = \theta^c$ and

$$h^b_{s_i+j-1} = h^c_{j|i}, \quad \forall j \in \mathcal{N}_0, \quad \forall i \in \mathcal{S}. \tag{19}$$

3. The unconstrained case:

$$R(H) = 0. \tag{20}$$

Here, we do not enforce any additional constraints on the hidden state sequences contained in $H$ (except compliance with the model equations (16b) and (16c)). We denote the solution to (16) under (20) as $(H^\infty, Y^\infty, \theta^\infty)$.

**Remark 3.** *Without loss of generality, we assume that the compact set $\mathcal{Y}$ used in (16d) is large enough such that $y_j(0, \theta^\infty, X_i^d) \in \mathcal{Y}$ and $y_j(0, \theta^c, X_i^d) \in \mathcal{Y}$ for all $j \in \mathcal{N}$ and all $i \in \mathcal{S}$, and furthermore, $y_t(0, \theta^*, X^d) \in \mathcal{Y}$ and $y_t(h^c_{0|1}, \theta^c, X^d) \in \mathcal{Y}$ for all times $t \in [1, T]$.*

## A.2 THE OUTPUT TURNPIKE PROPERTY

In this section, we establish a relation between $(H^*, Y^*, \theta^*)$ and the unconstrained solution $(H^\infty, Y^\infty, \theta^\infty)$. Note that this is an intermediate step which enables us to derive the desired relation between $(H^*, Y^*, \theta^*)$ and the coupled solution $(H^c, Y^c, \theta^c)$ in Sections A.3 and A.4. Our analysis is inspired by classical turnpike theory, which is an important concept in the context of optimal control (Trélat & Zuazua, 2025; Faulwasser & Grüne, 2022; Zaslavski, 2006; Carlson et al., 1991). In particular, we prove a turnpike property of the optimization problem (16), characterized in the output space. For the sake of clarity, we provide the following roadmap:

1. We first establish an upper bound on the difference of the value functions resulting from solving (16) under (17) and (20) (see Lemma 1). When interpreting the parameter $\theta$ as the (constant) control input to the dynamical system (1), this result can be interpreted as a stabilizability property of the latter, characterized in terms of the cost function (15): When increasing the prediction horizon $N$, the bound on the cost difference stays the same, which implies that the optimal input $\theta^*$ effectively stabilizes the "output reference" $Y^\infty$.

2. In Lemma 2, we establish a structural relation between the outputs $Y^*$ and $Y^\infty$ using the fact that both correspond to optimal solutions of the same optimization problem (16).

3. This result is used in Lemma 3 to prove a coercivity property of the difference of the value functions resulting from solving (16) under (17) and (20): The difference is positive definite in the sum of the differences of the corresponding output predictions.

4. Finally, by employing Lemmas 1 and 3, we can characterize the desired turnpike property of the problem (16) (see Proposition 1), which states that $Y^*$ must be close to $Y^\infty$ (i.e., the output turnpike) for most of the time. We provide a detailed interpretation of this result below Proposition 1.

**Lemma 1** (Stabilizability). *Let Assumption 1 hold. Then, there exists a constant $\bar{C} > 0$ such that*

$$\sum_{i=1}^{S} \sum_{j=m+1}^{N} l_{j|i}(y_{j|i}^*) - l_{j|i}(y_{j|i}^\infty) \leq \bar{C} S \lambda^m, \tag{21}$$

*where $\lambda \in (0,1)$ is from Assumption 1.*

*Proof.* The claim follows from the fact that $\theta^\infty \in \Theta$ by (16e) and Assumption 1. In particular, let us consider the candidate solution $(\hat{H}, \hat{Y}, \theta^\infty)$ such that $\hat{y}_{j|i} = y_j(0, \theta^\infty, X_i^{\mathrm{d}})$ for all $j \in \mathcal{N}$ and $i \in \mathcal{S}$. Here, $\hat{Y}$ contains $S$ output sequences of length $N$, each of which is generated by the RNN with $\theta^\infty$, initialized at zero, and being driven by the respective subsequences of the training input data $X_i^{\mathrm{d}}$, $i \in \mathcal{S}$. Note also that, without loss of generality, we can assume that the compact set $\mathcal{Y}$ is such that $\hat{y}_{j|i} \in \mathcal{Y}$ for all $j \in \mathcal{N}$ and all $i \in \mathcal{S}$, compare Remark 3. Consequently, $(\hat{H}, \hat{Y}, \theta^\infty)$ satisfies the constraints (16b)–(16f) and (17) and thus constitutes a feasible candidate solution to the problem (16) under (17). By optimality of $Y^*$, we have that $J_S(Y^*; D) \leq J_S(\hat{Y}; D)$, which yields

$$\sum_{i=1}^{S} \sum_{j=m+1}^{N} l_{j|i}(y_{j|i}^*) - l_{j|i}(y_{j|i}^\infty) \leq \sum_{i=1}^{S} \sum_{j=m+1}^{N} l_{j|i}(\hat{y}_{j|i}) - l_{j|i}(y_{j|i}^\infty). \tag{22}$$

Because both $\hat{y}_{j|i}$ and $y_{j|i}^\infty$ are elements of the compact set $\mathcal{Y}$ for each $j \in \mathcal{N}$ and $i \in \mathcal{S}$ and the stage cost $l_{j|i}(y)$ is quadratic in $y$, there exists a Lipschitz constant $L_l > 0$ such that

$$l_{j|i}(\hat{y}_{j|i}) - l_{j|i}(y_{j|i}^\infty) \leq \|l_{j|i}(\hat{y}_{j|i}) - l_{j|i}(y_{j|i}^\infty)\| \leq L_l \|\hat{y}_{j|i} - y_{j|i}^\infty\| \tag{23}$$

uniformly for all $\hat{y}_{j|i}, y_{j|i}^\infty \in \mathcal{Y}$ and all $j \in \mathcal{N}$ and $i \in \mathcal{S}$. (In fact, $L_l = 2(\max_{y \in \mathcal{Y}} \|y\| + \max_{y \in \mathcal{Y}^{\mathrm{d}}} \|y\|)$ is a valid choice, recalling that $\mathcal{Y}$ and $\mathcal{Y}^d$ are compact). Moreover, due to the fact that the output predictions $\hat{y}_{j|i}$ and $y_{j|i}^\infty$ are generated using the RNN with the same parameter $\theta^\infty \in \Theta$ but different initial hidden states, we can apply the property in (12) from Assumption 1 for each $i \in \mathcal{S}$ using $h_0^{(1)} = 0$ and $h_0^{(2)} = h_{0|i}^\infty$, which lets us infer that

$$\|\hat{y}_{j|i} - y_{j|i}^\infty\| \leq C\lambda^j \|h_{0|i}^\infty\| \leq C\bar{h}\lambda^j, \; j \in \mathcal{N}, \; i \in \mathcal{S}, \tag{24}$$

where $\bar{h} := \max_{h \in \mathcal{H}} \|h\|$ using (16d) and the fact that $\mathcal{H}$ is compact. The combination of (22)–(24) together with the application of the geometric series then leads to

$$\sum_{i=1}^{S} \sum_{j=m+1}^{N} l_{j|i}(y_{j|i}^*) - l_{j|i}(y_{j|i}^\infty)$$

$$\leq L_l C\bar{h} \sum_{i=1}^{S} \sum_{j=m+1}^{N} \lambda^j = L_l C\bar{h} \sum_{i=1}^{S} \lambda^m \frac{\lambda - \lambda^{N-m+1}}{1 - \lambda} \leq L_l C\bar{h} \sum_{i=1}^{S} \lambda^m \frac{\lambda}{1 - \lambda}.$$

By defining $\bar{C} := L_l C\bar{h}\lambda/(1 - \lambda)$ we obtain (21), which finishes this proof. $\square$

**Lemma 2.** *Consider the solutions $(H^*, Y^*, \theta^*)$ and $(H^\infty, Y^\infty, \theta^\infty)$. There exists a constant $\epsilon \geq 1$ such that*

$$\sum_{i=1}^{S} \sum_{j=m+1}^{N} 2(y_{j|i}^\infty - y_{j|i}^{\mathrm{d}})^\top (y_{j|i}^* - y_{j|i}^\infty) \geq -\frac{1}{\epsilon} \sum_{i=1}^{S} \sum_{j=m+1}^{N} \|y_{j|i}^* - y_{j|i}^\infty\|^2. \tag{25}$$

*Proof.* The property in (25) follows from optimality of $(H^*, Y^*, \theta^*)$ and $(H^\infty, Y^\infty, \theta^\infty)$. In particular, since $(H^*, Y^*, \theta^*)$ represents a feasible candidate solution to the unconstrained problem (16) under (20), we know that $J_S(Y^\infty) \leq J_S(Y^*)$, which leads to

$$\sum_{i=1}^{S} \sum_{j=m+1}^{N} l_{j|i}(y_{j|i}^*) - l_{j|i}(y_{j|i}^\infty) \geq 0. \tag{26}$$

Due to the fact that $l_{j|i}(y)$ is quadratic in $y \in \mathcal{Y}$, it is equivalent to its second-order Taylor polynomial. Consequently, we can express $l_{j|i}(y_{j|i}^*)$ in terms of $l_{j|i}(y_{j|i}^\infty)$ as

$$l_{j|i}(y_{j|i}^*) = l_{j|i}(y_{j|i}^\infty) + \frac{\partial l_{j|i}}{\partial y}(y_{j|i}^\infty)^\top (y_{j|i}^* - y_{j|i}^\infty) + \frac{1}{2}(y_{j|i}^* - y_{j|i}^\infty)^\top \mathfrak{H}(y_{j|i}^* - y_{j|i}^\infty)$$
$$= l_{j|i}(y_{j|i}^\infty) + 2(y_{j|i}^\infty - y_{j|i}^{\mathrm{d}})^\top (y_{j|i}^* - y_{j|i}^\infty) + \|y_{j|i}^* - y_{j|i}^\infty\|^2,$$

where $\mathfrak{H}$ is the Hessian of $l_{j|i}$ with respect to $y \in \mathcal{Y}$ and simply reduces to $\mathfrak{H} = 2I_p$. Thus, the left-hand side of (26) can be re-written as

$$\sum_{i=1}^{S} \sum_{j=m+1}^{N} l_{j|i}(y_{j|i}^*) - l_{j|i}(y_{j|i}^\infty) = \sum_{i=1}^{S} \sum_{j=m+1}^{N} \left( \|y_{j|i}^* - y_{j|i}^\infty\|^2 + 2(y_{j|i}^\infty - y_{j|i}^{\mathrm{d}})^\top (y_{j|i}^* - y_{j|i}^\infty) \right).$$
(27)

The inequality in (26) then implies that

$$\sum_{i=1}^{S} \sum_{j=m+1}^{N} 2(y_{j|i}^\infty - y_{j|i}^{\mathrm{d}})^\top (y_{j|i}^* - y_{j|i}^\infty) \geq -\sum_{i=1}^{S} \sum_{j=m+1}^{N} \|y_{j|i}^* - y_{j|i}^\infty\|^2,$$

which establishes the desired property in (25) for $\epsilon = 1$ and hence concludes this proof. $\qquad \square$

**Lemma 3** (Coercivity). *Let the property in* (25) *hold for some $\epsilon > 1$. Then, the solutions $(H^*, Y^*, \theta^*)$ and $(H^\infty, Y^\infty, \theta^\infty)$ satisfy*

$$\sum_{i=1}^{S} \sum_{j=m+1}^{N} l_{j|i}(y_{j|i}^*) - l_{j|i}(y_{j|i}^\infty) \geq \frac{\epsilon - 1}{\epsilon} \sum_{i=1}^{S} \sum_{j=m+1}^{N} \|y_{j|i}^* - y_{j|i}^\infty\|^2.$$
(28)

*Proof.* The claim follows by applying the property (25) to (27) in the proof of Lemma 2. $\qquad \square$

**Remark 4.** *Requiring $\epsilon > 1$ in Lemma 3 is a rather mild technical condition on $Y^*$ with respect to $Y^\infty$. In particular, using $\kappa := 1/\epsilon \in (0,1)$, the result fails only if, for $\sum_{i=1}^{S} \sum_{j=m+1}^{N} \|y_{j|i}^* - y_{j|i}^\infty\|^2 \neq 0$, the following holds:*

$$-1 \leq \frac{\sum_{i=1}^{S} \sum_{j=m+1}^{N} 2(y_{j|i}^\infty - y_{j|i}^{\mathrm{d}})^\top (y_{j|i}^* - y_{j|i}^\infty)}{\sum_{i=1}^{S} \sum_{j=m+1}^{N} \|y_{j|i}^* - y_{j|i}^\infty\|^2} < -\kappa, \quad \forall \kappa \in (0,1),$$

*which is the case if and only if*

$$\sum_{i=1}^{S} \sum_{j=m+1}^{N} 2(y_{j|i}^\infty - y_{j|i}^{\mathrm{d}})^\top (y_{j|i}^* - y_{j|i}^\infty) = -\sum_{i=1}^{S} \sum_{j=m+1}^{N} \|y_{j|i}^* - y_{j|i}^\infty\|^2 \neq 0.$$
(29)

*Solutions to this equation correspond to the case where $J_S(Y^*; D) = J_S(Y^\infty; D)$ but $\|y_{j|i}^* - y_{j|i}^\infty\|^2 \neq 0$ for some $j \in \mathcal{N}$ with $j \geq m+1$ and $i \in \mathcal{S}$. This may occur if $Y^\infty$ (specifically, its restriction to the interval $[m+1, N]$) is non-unique; in particular, the above implies that $Y^*$ then also corresponds to an optimal solution to the unconstrained problem. But this means that we can simply set $(H^\infty, Y^\infty, \theta^\infty) = (H^*, Y^*, \theta^*)$, rendering condition (25) again valid for any $\epsilon > 1$. Alternatively, (29) could be avoided in advance by proper design of the RNN, ensuring that the output turnpike $Y^\infty$ is unique (which does* not *require uniqueness of the solution with respect to the hidden state trajectory $H^\infty$ or $\theta^\infty$, compare also the discussion below Proposition 1). Moreover, we also conjecture that our analysis can be extended to explicitly cover the case where $Y^\infty$ is non-unique using arguments inspired by Trélat & Zhang (2018); namely, by characterizing the turnpike property using a turnpike set $\mathcal{T}$ containing all possible output sequences achieving the minimum cost (instead of one particular output sequence) and by deriving the results in terms of the point-to-set distance of the solution $Y^*$ to the turnpike set $\mathcal{T}$. Finally, we would like to point out that in our synthetic experiment conducted in Section 5.1, where we were able to explicitly check this condition, it was always possible to choose $\epsilon > 1$ such that property (25) is satisfied (even when changing the network architecture and using different nonlinear activation functions).*

**Proposition 1** (Output turnpike property). *Let Assumption 1 be satisfied. Consider the solutions $(H^*, Y^*, \theta^*)$ and $(H^\infty, Y^\infty, \theta^\infty)$ and suppose that the property in (25) holds for some $\epsilon > 1$. Then, there exists a constant $K > 0$ such that*

$$\sum_{j=m+1}^{N} \frac{1}{S} \sum_{i=1}^{S} \|y_{j|i}^* - y_{j|i}^\infty\|^2 \le K\lambda^m, \tag{30}$$

*where $\lambda \in (0,1)$ is from Assumption 1.*

*Proof.* The statement follows by considering the difference of the loss functions

$$\sum_{i=1}^{S} \sum_{j=m+1}^{N} l_{j|i}(y_{j|i}^*) - l_{j|i}(y_{j|i}^\infty)$$

and applying the lower bound from Lemma 3 and the upper bound from Lemma 1. The definition of $K := \bar{C}\epsilon/(\epsilon - 1)$ with $\bar{C}$ from Lemma 1 then leads to (30), which finishes this proof. $\qquad\square$

Some remarks are in order.

- Proposition 1 provides a characterization of the $j$-step prediction error, averaged over all $S$ subsequences:

$$e_j^\infty := \frac{1}{S} \sum_{i=1}^{S} \|y_{j|i}^* - y_{j|i}^\infty\|^2, \quad j \in \mathcal{N}.$$

  Specifically, property (30) implies that the sum of $e_j^\infty$ over $j \in [m+1, N]$ is uniformly bounded *independent of the length of the subsequences $N$*. For a fixed value of $m$, this must imply that $e_j^\infty$ must become smaller for larger values of $N$ for most of the times $j \in [m+1, N]$. By non-negativity of the norm, we can conclude a similar statement for the output predictions corresponding to each subsequence $i \in \mathcal{S}$: $y_{j|i}$ gets closer to $y_{j|i}^\infty$ for most of the times $j \in [m+1, N]$ when increasing the value of $N$. Hence, the latter can be regarded as the turnpike for the former, and the larger we choose the length $N$, the more elements of $y_{j|i}$ are close to $y_{j|i}^\infty$.

- The fact that the right-hand side of (30) is independent of $N$ is in line with intuition: Differences in the output predictions (and thus in the loss functions of the optimization problems under consideration) are caused solely by the burn-in phase of the RNN resulting from zero initialization, whose influence is limited and, for a fixed network parameterization $\theta$, does not depend on $N$.

- The main difference to the turnpike-related literature is that we formulate the turnpike property with respect to the *outputs*—without, as usual, characterizing the behavior of optimal states, controls, and possibly adjoints, compare Trélat & Zuazua (2025); Faulwasser & Grüne (2022); Zaslavski (2006); Carlson et al. (1991). This is natural in our setting, as the set of hidden states and network parameters (i.e., the "controls") that generates a specific output sequence is typically *not* a singleton (the same output sequence can usually be generated by a variety of hidden states and parameters without significantly restricting the network architecture).

- The turnpike characterization in (30) is conceptually related to the *interval turnpike property* from Faulwasser et al. (2022). Note that (30) can be easily reformulated in terms of a *measure/cardinality turnpike property* by estimating the number of elements of $Y^*$ that lie outside a certain $\epsilon$-neighborhood of the turnpike $Y^\infty$. Similar characterizations are prominent in the most recent turnpike-related literature (Trélat & Zuazua, 2025; Faulwasser & Grüne, 2022; Grüne & Pirkelmann, 2019; Faulwasser et al., 2018); however, the quantitative nature of the bound in (30) turns out to be more useful when establishing the results in Sections A.3 and A.4 below.

- The fact that we establish the turnpike property (Proposition 1) using stabilizability (Lemma 1) and coercivity (Lemma 3) is in line with standard turnpike results (compare, e.g., Trélat & Zuazua (2025); Trélat & Zhang (2018); Grüne & Müller (2016)). There,

however, coercivity is usually first established as a consequence of an additional dissipativity assumption on the problem, mainly due to the fact that the results are tailored to general cost functions. In contrast, here we can leverage the quadratic nature of the MSE loss and thus avoid an additional dissipativity condition. Nevertheless, we would like to point out that, as one might expect, Lemma 3 directly implies a dissipativity property, where the storage function can be chosen as a constant and the supply rate involves a strictness term capturing the output differences.

### A.3 ACCURACY AND REGRET OVER SUBSEQUENCES OF THE TRAINING DATA

In this section, we leverage the turnpike property provided by Proposition 1 to establish accuracy and regret guarantees for the solution $(H^*, Y^*, \theta^*)$ with respect to the benchmark $(H^c, Y^c, \theta^c)$. The following should be noted in this regard.

**Remark 5.** *The previous results (in particular, Lemmas 1–3 and Proposition 1) also hold true if $(H^*, Y^*, \theta^*)$ in the statement of the results is replaced by the coupled solution $(H^c, Y^c, \theta^c)$. The respective proofs can be straightforwardly adapted using the fact that $(H^c, Y^c, \theta^c)$ is a solution to the optimization problem* (16).

Hence, by combining Proposition 1 and Remark 5, we can characterize the relation between $(H^*, Y^*, \theta^*)$ and $(H^c, Y^c, \theta^c)$ as shown in the following corollary.

**Corollary 1.** *Let Assumption 1 be satisfied. Suppose that there exists some $\epsilon > 1$ small enough such that the property in* (25) *holds true for both $(H^*, Y^*, \theta^*)$ and $(H^c, Y^c, \theta^c)$, see Remark 5. Then, the following statements hold:*

1. *Output accuracy: The learned output sequences contained in $Y^*$ and $Y^c$ satisfy*

$$\sum_{j=m+1}^{N} \frac{1}{S} \sum_{i=1}^{S} \|y_{j|i}^* - y_{j|i}^c\|^2 \leq 4K\lambda^m, \tag{31}$$

   *where $K > 0$ is from Proposition 1 and $\lambda \in (0, 1)$ is from Assumption 1.*

2. *Training regret: $Y^*$ and $Y^c$ satisfy*

$$J_S(Y^*; D) - J_S(Y^c; D) \leq \bar{C} \cdot \frac{\lambda^m}{N - m}, \tag{32}$$

   *where $\bar{C} > 0$ is from Lemma 1 and $\lambda \in (0, 1)$ is from Assumption 1.*

*Proof.* The first statement is a consequence of Proposition 1, Remark 5, and application of the Cauchy-Schwartz and Young's inequality. In particular, (31) follows by noting that

$$\sum_{j=m+1}^{N} \frac{1}{S} \sum_{i=1}^{S} \|y_{j|i}^* - y_{j|i}^c\|^2$$

$$\leq 2 \sum_{j=m+1}^{N} \frac{1}{S} \sum_{i=1}^{S} \|y_{j|i}^* - y_{j|i}^\infty\|^2 + 2 \sum_{j=m+1}^{N} \frac{1}{S} \sum_{i=1}^{S} \|y_{j|i}^c - y_{j|i}^\infty\|^2$$

$$\leq 4K\lambda^m.$$

The second statement can be established by applying similar steps as in the proof of Lemma 1. In particular, using the candidate solution $\hat{y}_{j|i} = y_j(0, \theta^c, X_i^d) \in \mathcal{Y}$, $j \in \mathcal{N}$, $i \in \mathcal{S}$, we have that $J_S(Y^*; D) - J_S(Y^c; D) \leq J_S(\hat{Y}; D) - J_S(Y^c; D)$ by optimality of $Y^*$. Now, we can apply the same arguments that followed (22), which leads to (32) and thus concludes this proof. □

**Remark 6.** *The estimate on the training regret in* (32) *in fact also holds true for $\epsilon \geq 1$, as it does not require the coercivity property established in Lemma 3.*

Recalling the equivalence between the coupled solution $(H^c, Y^c, \theta^c)$ and the actual benchmark consisting of the hidden state sequence $\{h_t^b\}_{t=0}^T$ and the parameter $\theta^b$ as specified in (19), one can directly see that Corollary 1 formalizes (and thus proves) Theorem 1.

### A.4 ACCURACY AND REGRET OVER THE FULL TRAINING DATA

In this section, we establish accuracy and regret estimates for the RNN parameterized by $\theta^*$ when processing the full training input sequence $X^{\mathrm{d}}$. The estimates are given with respect to the accuracy achieved by the benchmark $(h_0^{\mathrm{b}}, \theta^{\mathrm{b}})$ (that is, the the solution to (11)), where we measure the performance using the truncated MSE loss as defined in (9).

The following result constitutes the formal version of Theorem 2.

**Theorem 3.** *Let Assumption 1 be satisfied. Suppose that there exists some $\epsilon > 1$ small enough such that the property in (25) holds true for both $(H^*, Y^*, \theta^*)$ and $(H^{\mathrm{c}}, Y^{\mathrm{c}}, \theta^{\mathrm{c}})$, cf. Remark 5. Then, the following statements hold:*

1. *Averaged output accuracy: There exists a constant $E_1 > 0$ such that*

$$\frac{1}{T-m} \sum_{t=m+1}^{T} \|y_t(0, \theta^*, X^{\mathrm{d}}) - y_t(h_0^{\mathrm{b}}, \theta^{\mathrm{b}}, X^{\mathrm{d}})\|^2 \leq E_1 \cdot \frac{(S-1)\lambda^{2o_{\min}} + S\lambda^m}{T-m} \quad (33)$$

2. *Performance regret: There exists a constant $E_2 > 0$ such that*

$$P(0, \theta^*; D) - P(h_0^{\mathrm{b}}, \theta^{\mathrm{b}}; D) \leq E_2 \cdot \sqrt{\frac{(S-1)\lambda^{2o_{\min}} + S\lambda^m}{T-m}}. \quad (34)$$

*Proof.* We start with some useful definitions and observations. Let $\{h_t^*\}_{t=0}^{T}$ be the hidden state sequence that is generated by the RNN from (1)–(2) parameterized by $\theta^*$ when being evaluated over the full training input sequence $X^{\mathrm{d}}$ and initialized with $h_0 = 0$. Define the corresponding output sequence as $y_t^* := y_t(0, \theta^*, X^{\mathrm{d}})$, $t \in [1, T]$. Similarly, define the benchmark output sequence $y_t^{\mathrm{b}} = y_t(h_0^{\mathrm{b}}, \theta^{\mathrm{b}}, X^{\mathrm{d}})$, $t \in [1, T]$. Recall that $(h_0^{\mathrm{b}}, \theta^{\mathrm{b}}) = (h_{0|1}^{\mathrm{c}}, \theta^{\mathrm{c}})$ by (19), which yields

$$y_{s_i+j-1}^{\mathrm{b}} = y_{j|i}^{\mathrm{c}} \quad \forall j \in \mathcal{N}, \quad \forall i \in \mathcal{S}. \quad (35)$$

Without loss of generality, we can assume that $h_t^* \in \mathcal{H}$ and $y_t^*, y_t^{\mathrm{b}} \in \mathcal{Y}$ for all $t \in [1, T]$, compare Remark 3.

We proceed to prove the first claim of this theorem. To this end, note that we can re-write the sum of output differences (i.e., the left-hand side of (33)) in terms of the subsequences $i \in \mathcal{S}$ using the definition of the overlap $o_i$ from (5) according to

$$\sum_{t=m+1}^{T} \|y_t^* - y_t^{\mathrm{b}}\|^2 = \sum_{j=m+1}^{N} \|y_j^* - y_j^{\mathrm{b}}\|^2 + \sum_{i=2}^{S} \sum_{j=o_i+1}^{N} \|y_{s_i+j-1}^* - y_{s_i+j-1}^{\mathrm{b}}\|^2. \quad (36)$$

Regarding the first sum on the right-hand side of (36), each $y_j^*$ coincides with the output of the first training subsequence in the sense that $y_j^* = y_j(0, \theta^*, X^{\mathrm{d}}) = y_j(0, \theta^*, X_1^{\mathrm{d}}) = y_{j|1}^*$ for all $j \in \mathcal{N}$. Using the Cauchy-Schwarz and Young's inequality together with (35), it follows that

$$\|y_{s_i+j-1}^* - y_{s_i+j-1}^{\mathrm{b}}\|^2 = \|y_{s_i+j-1}^* - y_{j|i}^* + y_{j|i}^* - y_{s_i+j-1}^{\mathrm{b}}\|^2$$
$$\leq 2\|y_{j|i}^* - y_{j|i}^{\mathrm{c}}\|^2 + 2\|y_{s_i+j-1}^* - y_{j|i}^*\|^2$$

for all $j \in \mathcal{N}$ and $i \in \mathcal{S}$. Consequently, (36) can be bounded as

$$\sum_{t=m+1}^{T} \|y_t^* - y_t^{\mathrm{b}}\|^2 \leq \sum_{j=m+1}^{N} \|y_{j|1}^* - y_{j|1}^{\mathrm{c}}\|^2 + 2 \sum_{i=2}^{S} \sum_{j=o_i+1}^{N} \|y_{j|i}^* - y_{j|i}^{\mathrm{c}}\|^2$$
$$+ 2 \sum_{i=2}^{S} \sum_{j=o_i+1}^{N} \|y_{s_i+j-1}^* - y_{j|i}^*\|^2. \quad (37)$$

We first focus on the last sum on the right-hand side in (37). Here, each summand corresponds to the difference of network outputs resulting from different initial hidden states, but using the same

parameter $\theta^*$ which satisfies Assumption 1 by (16e). In particular, for each $i \in \mathcal{S}$ with $i \geq 2$, from (12) we obtain

$$
\begin{aligned}
\|y^*_{s_i+j-1} - y^*_{j|i}\|^2 &= \|y_{s_i+j-1}(0, \theta^*, X^{\mathrm{d}}) - y_j(0, \theta^*, X^{\mathrm{d}}_i)\|^2 \\
&= \|y_j(h^*_{s_i-1}, \theta^*, X^{\mathrm{d}}_i) - y_j(0, \theta^*, X^{\mathrm{d}}_i)\|^2 \\
&\leq C^2 \lambda^{2j} \|h^*_{s_i-1}\|^2 \leq c_1 \lambda^{2j}
\end{aligned}
$$

for all $j \in \mathcal{N}$, where we have used the definition of $c_1 := C^2 \max_{h \in \mathcal{H}} \|h\|^2$ (noting that $c_1 > 0$ is well-defined due to compactness of $\mathcal{H}$ and $h^*_t \in \mathcal{H}$ for all $t \in [0, T]$, see above). By summing over $i, j$ and applying the geometric series, it follows that

$$
\sum_{i=2}^{S} \sum_{j=o_i+1}^{N} \|y^*_{s_i+j-1} - y^*_{j|i}\|^2 \leq \sum_{i=2}^{S} \sum_{j=o_i+1}^{N} c_1 \lambda^{2j} = \sum_{i=2}^{S} c_1 \frac{\lambda^2}{1-\lambda^2} \lambda^{2o_i} \left(1 - \lambda^{2(N-o_i)}\right)
$$
$$
\leq c_2 \cdot (S-1) \lambda^{2o_{\min}}, \tag{38}
$$

where $c_2 := c_1 \lambda^2 / (1 - \lambda^2)$.

The first line of the right-hand side in (37) can be bounded using the fact that the burn-in phase length $m$ satisfies $m \leq o_{\min} \leq o_i$ for all $i \geq 2$ together with the turnpike property provided by Corollary 1, which leads to

$$
\sum_{j=m+1}^{N} \|y^*_{j|1} - y^{\mathrm{c}}_{j|1}\|^2 + 2 \sum_{i=2}^{S} \sum_{j=o_i+1}^{N} \|y^*_{j|i} - y^{\mathrm{c}}_{j|i}\|^2 \leq 2 \sum_{i=1}^{S} \sum_{j=m+1}^{N} \|y^*_{j|i} - y^{\mathrm{c}}_{j|i}\|^2
$$
$$
\leq 2 \cdot 4KS\lambda^m. \tag{39}
$$

The combination of (37)–(39) then yields

$$
\sum_{t=m+1}^{T} \|y^*_t - y^{\mathrm{b}}_t\|^2 \leq 2c_2(S-1)\lambda^{2o_{\min}} + 8SK\lambda^m.
$$

Using the definition $E_1 := 2 \cdot \max\{c_2, 4K\}$ and dividing both sides by $(T-m) > 0$, we obtain (33).

To prove the second statement, note that the performance criterion in (9) is Lipschitz on compact sets (recall out observations in the beginning of this proof, the arguments that lead to (23), and the definition of the stage cost in (14)); consequently, the left-hand side of (34) can be bounded according to

$$
\begin{aligned}
P(0, \theta^*; D) - P(h^{\mathrm{b}}_0, \theta^{\mathrm{b}}; D) &= \|P(0, \theta^*; D) - P(h^{\mathrm{b}}_0, \theta^{\mathrm{b}}; D)\| \\
&= \frac{1}{T-m} \sum_{t=m+1}^{T} \left| \|y^*_t - y^{\mathrm{d}}_t\|^2 - \|y^{\mathrm{b}}_t - y^{\mathrm{d}}_t\|^2 \right| \\
&\leq \frac{L_l}{T-m} \sum_{t=m+1}^{T} \|y^*_t - y^{\mathrm{b}}_t\|. \tag{40}
\end{aligned}
$$

Using concavity of the square root and Jensen's inequality, it holds that

$$
\sum_{t=m+1}^{T} \|y^*_t - y^{\mathrm{b}}_t\| = \sum_{t=m+1}^{T} \sqrt{\|y^*_t - y^{\mathrm{b}}_t\|^2} \leq \frac{T-m}{\sqrt{T-m}} \left(\sum_{t=m+1}^{T} \|y^*_t - y^{\mathrm{b}}_t\|^2\right)^{1/2} \tag{41}
$$

From (40), (41), and (33) proven above, we obtain

$$
\begin{aligned}
P(0, \theta^*; D) - P(h^{\mathrm{b}}_0, \theta^{\mathrm{b}}; D) &\leq \frac{L_l}{T-m} \frac{T-m}{\sqrt{T-m}} \left(E_1 \cdot \left((S-1)\lambda^{2o_{\min}} + S\lambda^m\right)\right)^{1/2} \\
&\leq L_l \sqrt{E_1} \sqrt{\frac{(S-1)\lambda^{2o_{\min}} + S\lambda^m}{T-m}}.
\end{aligned}
$$

This establishes (34) and hence finishes this proof. $\qquad \square$

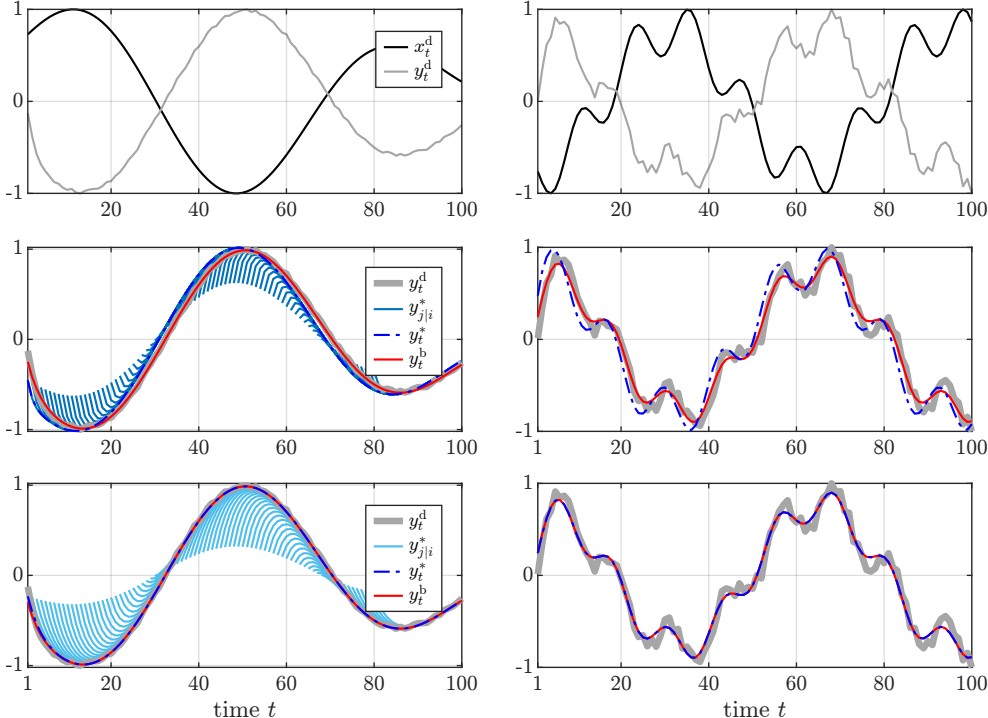

Figure 5: Supplementary results for the synthetic data experiment performed in Section 5.1. The left column refers to the training data, the right column to the test data. The corresponding input-output time series are shown in the top row. In the middle row, we compare the outputs of the RNN trained with $m = 0$ (without burn-in) in terms of the outputs $y_{j|i}^*$ (belonging to subsequences of the training data $X_i^{\mathrm{d}}$, $i \in \mathcal{S}$) and the outputs $y_t^*$ (generated by processing the entire input sequence $X^{\mathrm{d}}$) with the benchmark output $y_t^{\mathrm{b}}$ and the ground truth data $y_t^{\mathrm{d}}$. The bottom row shows the respective results when training the RNNs using the burn-in phase length $m = N - 1 = 20$.

# B  ADDITIONAL MATERIAL FOR THE EXPERIMENTS

## B.1  SYNTHETIC DATA

In this section, we provide additional illustrative figures and interpretations of the results of the synthetic experiment performed in Section 5.1.

Specifically, the left column in Figure 5 compares the generated outputs of the trained RNN, the benchmark RNN, and the ground truth data. Without using a burn-in phase (i.e., $m = 0$, middle row), the transient resulting from the zero initialization causes the output $y_t^* = y_t(0, \theta^*, X^{\mathrm{d}})$ to be relatively far away from the benchmark output $y_t^{\mathrm{b}} = y_t(h_0^{\mathrm{b}}, \theta^{\mathrm{b}}, X^{\mathrm{d}})$ (which employs optimal initialization, compare Section 4.1) and the ground truth data. This is because the optimizer tries to minimize the influence of this transient by minimizing the convergence rate, which comes at the expense of long-term accuracy. In contrast, when using a burn-in phase $m \geq 12$, this transient is effectively excluded from the cost function, letting the outputs $y_{j|i}^* = y_j(0, \theta^*, X_i^{\mathrm{d}})$ converge closely to the benchmark (and the ground truth data).

According to the right column in Figure 5, these observations directly carry over to performance of the trained RNN on test data. Consequently, tuning the burn-in phase during training enables the RNN to achieve benchmark performance on both the training and the test data.

## B.2  DATA SETS FROM SYSTEM IDENTIFICATION AND TIME SERIES FORECASTING

This section contains supplementary material on the experiments conducted in Section 5.2. Parameters depending on the respective data sets are reported in Table 2.

Table 2: Training parameters for the experiments performed in Section 5.2.

| data set | $T$ | $N$ | $n_y$ | $\alpha$ | $b$ | epochs | $T_{\text{test}}$ | $T_{\text{val}}$ |
|----------|-----|-----|-------|----------|-----|--------|-------------------|------------------|
| Silver-Box | 20,000 | 100..1000 | 1 | $10^{-4}$ | 100 | 400 | 10,000 | 10,000 |
| W-H | 20,000 | 100..1000 | 1 | $10^{-4}$ | 100 | 400 | 10,000 | 10,000 |
| RLC | 6667 | 100..1000 | 2 | $3\cdot10^{-4}$ | 100 | 1000 | 3333 | 3333 |
| Electricity (5) | 10,000 | 48, 96, 192 | 96 | $5\cdot10^{-5}$ | 24 | 400 | 3360 | 3360 |
| Traffic (9) | 10,000 | 48, 96, 192 | 96 | $5\cdot10^{-5}$ | 24 | 400 | 3360 | 3360 |
| Solar (2) | 20,000 | 48, 96, 192 | 96 | $2\cdot10^{-4}$ | 24 | 300 | 5000 | 5000 |

The variables correspond to the training sequence length $T$, the TBPTT subsequence (lookback window) length $N$, the RNN output dimension $n_y$, the step size $\alpha$ used in the Adam algorithm, the batch size $b$, the test sequence length $T_{\text{test}}$, and the validation sequence length $T_{\text{val}}$. The values in parentheses indicate the considered variate of the respective time series data (selected randomly).

For all experiments, we divide the training sequence of length $T$ into $S = T - N + 1$ subsequences of length $N$, leading to the one-step overlap $o_i = 1$ for all $i \in \mathcal{S}$ with $i \geq 2$, compare Section 3.1. This allows us leverage the maximum number of subsequences during TBPTT training and ultimately ensure a fair comparison between the considered methods and parameter choices. A thorough comparison of various options for the number of subsequences and how they might overlap is out of the scope of this paper, as this strongly depends on availability of additional prior knowledge and expertise about the data sets in order to decide which sequences actually contain information and are important for training, and which do not. However, this might indeed be relevant in certain scenarios (especially for very long training sequences), where a proper selection of the subsequences used for training (and hence the value of $S$) can potentially significantly accelerate the training procedure without sacrificing overall performance.

We trained the RNNs in a standard Python environment using PyTorch and the Adam optimizer (Kingma & Ba, 2015). Computations are performed on a standard laptop (Intel Core Ultra 7 265H, 32 GB RAM, NVIDIA RTX PRO 500 Blackwell GPU).

For the training of RNNs for time series forecasting applied to the Electricity, Traffic, and Solar-Energy data sets, we first split the available time series into a training segment of length $T$ and a test segment of length $T_{\text{test}}$. Then, we pre-train the RNNs on a subsequence of the training data of length $T - T_{\text{val}}$ using different values of $N$ and $m$. The remaining subsequence of length $T_{\text{val}}$ is used for evaluating the pre-training performance and ultimately selecting $m^*$ based on the MSE achieved during pre-training and validation. Then, for every $N$ and $m$, we re-train the RNN using the full training sequence of length $T$, and evaluate the forecast performance (and hence our choice of $m^*$) on the test segment of length $T_{\text{test}}$. Stateful TBPTT and BPTT training is performed directly using the full training sequence of length $T$.

In Figure 6, we compare the MSE achieved by each RNN after TBPTT training for various combinations of the window length $N$ and burn in phase $m$. The top row corresponds to system identification datasets, and the bottom row to time series forecasting data sets. For the latter, we show the training and test MSE achieved after re-training the RNNs on the full training sequence of length $T$ using the previously selected values of $N$ and $m$.

In all experiments, we find that the performance achieved on both the training and test data is influenced by the choices of $N$ and $m$, as suggested by our theoretical results. Moreover, proper tuning of $N$ leads to a reduction of the MSE achieved on the training and test data of more than 60% in some cases, see Table 1.

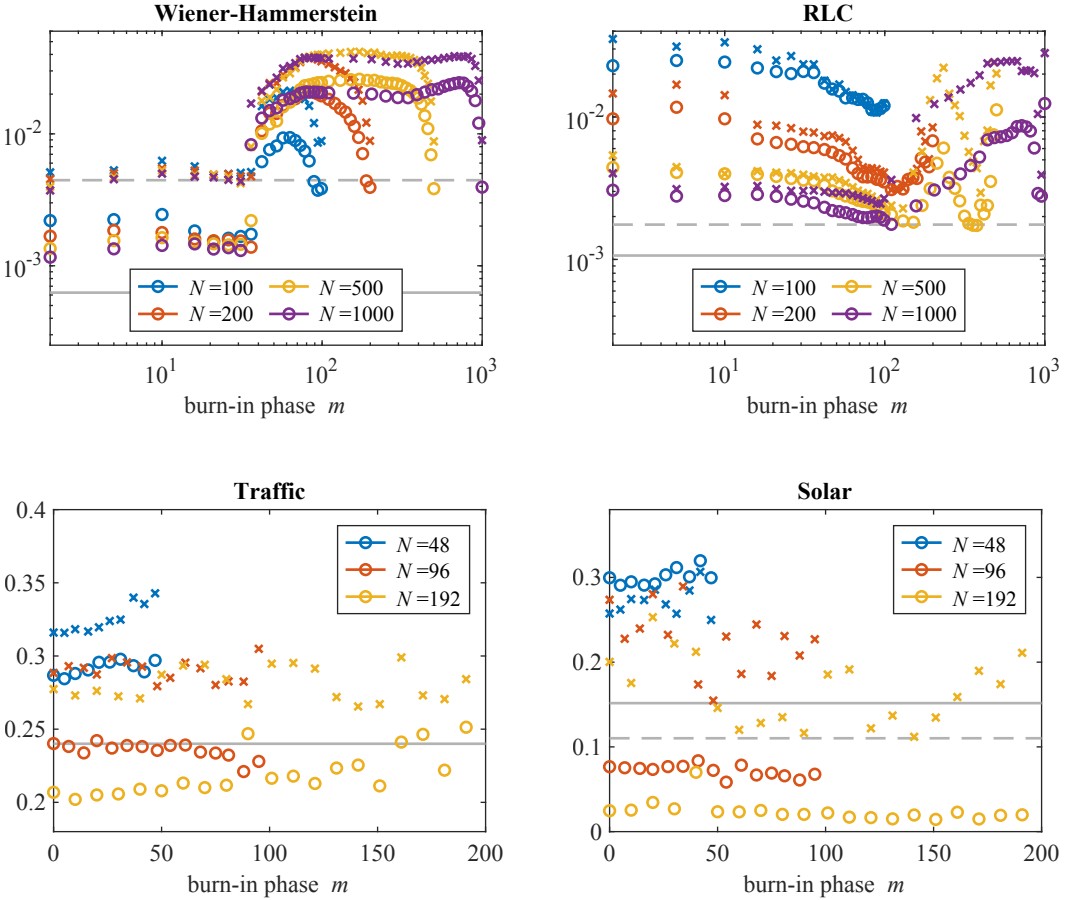

Figure 6: MSE after (re-)training the respective RNNs using TBPTT and various combinations of the window length $N$ and burn-in phase $m$; '∘'-markers correspond to the MSE achieved on the training data, '×'-markers to the MSE on the test data. Gray lines refer to the BPTT-trained RNN evaluated on the training (solid) and test data (dashed). This figure complements Figure 4.

