# OpenReview forum: "Tuning the burn-in phase in training recurrent neural networks improves their performance"
_ICLR.cc/2026/Conference — ICLR 2026 Poster_

### Official Review · Reviewer_xtPM · 2025-10-24

**Soundness:** 3
**Presentation:** 3
**Contribution:** 1
**Rating:** 4
**Confidence:** 2

**Summary:**

This paper assesses the effects of training recurrent neural networks using truncated back-propagation through time (TBPTT). In TBPTT the full training sequence is split into a number of overlapping sub-sequences to which learning is applied. Each sub-sequence includes an initial part (the burn-in phase) that is excluded from the loss. This paper seeks to access the loss in performance produced by approximating BPTT with TBPTT, and the effects of the burn-in period on performance.

**Strengths:**

The paper combines both theoretical and empirical research.

It well structured and generally clearly written.

**Weaknesses:**

The theoretical and practical consequences of this work are unclear.

The most detailed empirical results are provided for a synthetic task that is so simple it is unlikely to have any practical relevance.

**Questions:**

For all the empirical experiments, why not provide results for BPTT to provide a baseline to compare to?

For the three real-world data-sets why not perform experiments for a range of different combinations of m and N?

Can you quantify how much TBPTT reduces computational costs? e.g. What is the training time for each method that you compare?

Given that different hyper-parameters are required to obtain good performance with TBPTT on different tasks, how could hyper-parameter tuning be performed efficiently? Is the time taken to perform hyper-parameter tuning smaller than the time it would take to perform BPTT?

Hyper-parameter tuning is a particular concern as the optimal hyper-parameters found for the training data do not seem to produce the best results on the testing data. It would therefore be particularly informative to compare the test-data performance of BPTT with that of TBPTT using the hyper-parameters optimised on the training data.

---

> ### Author Response · Authors · 2025-11-24
> **Response to Reviewer xtPM (Part 1/2)**
>
> We thank the reviewer very much for careful review of our paper as well as providing valuable suggestions and constructive criticism. We have revised the manuscript accordingly, and we believe that the changes significantly contribute to the overall quality and relevance of our paper. Please see our detailed answers below.
>
>
>
> ## **W1: Theoretical and practical consequences**
>
> Thank you for pointing out this lack of clarity. Based on our theoretical results and the newly obtained empirical findings, our work provides the following two main contributions in the context of RNN theory and practice:
>
>   1. We develop the concept of burn-in phase in TBPTT training of RNNs from a heuristic to a theoretically grounded method.
>   2. Tuning the burn-in phase can yield significant performance increase.
>
>   In the revised version of our manuscript, we have added a paragraph at the end of Section 1 in this regard. This contributes to the overall structure of our work and clearly presents the most important findings for the reader.
>
>
>
> ## **W2: Practical relevance of the results**
>
> We thank the reviewer for pointing out the lack of practical relevance. We have significantly extended our experiments in several ways. In particular:
>
>   1. We have considered various combinations of TBPTT window length *N* and burn-in phase *m*. For more details, see Q2.
>   2. We have compared the training results to additional baseline methods: full BPTT and stateful TBPTT (where hidden states are passed over subsequences during training), compare our answer to Q1 below.
>   3. We have made tuning of *m* more systematic using additional validation data sets, see Q4/Q5 below.
>   4. We have successfully tested the trained TBPTT-trained RNNs for compliance with Assumption 1, which is the main requirement of our theoretical results to be valid.
>
>   In all experiments, we observe a strong influence of the burn-in phase on the training process, and proper tuning can lead to a reduction of the MSE achieved on the training and test data of more than 60% in some cases. To summarize, we believe that the changes we made in accordance with this reviewer's comments and the newly obtained empirical results significantly contribute to the practical relevance our our work.
>
>
>
> ## **Q1: BPTT baseline**
>
> Thank you for this suggestion! We agree with the reviewer that it is both interesting and important to compare TBPTT training results with BPTT. Hence, for all experiments, we now also provide BPTT training results in the revised version of our manuscript. Overall, we find that proper tuning of *m* in TBPTT narrows the gap to BPTT performance on the training data, while avoiding the risk of overfitting and reducing the training time, see Table 1.
>
>
>
> ## **Q2: Extended experiments**
>
> Thank you for this suggestion! We fully agree and extended our experiments accordingly. We now perform TBPTT training for various combinations of *N* and *m*, see Figures 4 and 6 and Table 1 in the revised version of our manuscript. As expected, the performance generally increases with increasing *N*, while training time also increases. Interestingly, we observe the same qualitative behavior when varying *m*, irrespective of the window length *N*.
>
>
> ## **Q3: Computational costs**
>
>
> Thank you for raising this question! Indeed, we agree that training times should be reported and compared to each other, especially with respect to the newly added BPTT training. To this end, we have included the right figure in Figure 4, where we now report the training times of TBPTT training for different window lengths *N* relative to the BPTT baseline. Here, we find that TBPTT training overall requires a fraction of BPTT, which was to be expected and is mainly due to the fact that computations can be parallelized.
>
> ## **Q4: Hyperparameter tuning**
>
> Thank you for these questions. Tuning the hyper-parameters (in particular, the value of *m*) can be done systematically as usual based on additional validation data sets. However, we agree that this may be a tedious task, depending on the respective application. Nevertheless, we want to emphasize that there are many scenarios, in which performing BPTT is actually not an option, and TBPTT is necessarily required. First, this applies to highly complex RNN architectures and large training sequences, which significantly complicate computing the required gradients during training, also because of vanishing/exploding gradients. Moreover, BPTT is naturally undesired in time series forecasting applications, where the forecasts are typically generated by considering a lookback window of fixed length, which directly correspond to the TBPTT window length *N* in our analysis.

---

> ### Author Response · Authors · 2025-11-24
> **Response to Reviewer xtPM (Part 2/2)**
>
> ## **Q5: Test performance after tuning**
>
> Thank you very much for this suggestion! We fully agree with your assessment. First, we have made the tuning of *m* more systematic by considering performance on the respective training and additional validation data sets. Moreover, from the newly obtained training results (see Table 1 in the revised version of our manuscript), we find that proper tuning of *m* in TBPTT narrows the gap to BPTT performance on the training (and test) data while avoiding the risk of overfitting and reducing the training time.

---

### Official Review · Reviewer_4peq · 2025-11-01

**Soundness:** 2
**Presentation:** 1
**Contribution:** 2
**Rating:** 4
**Confidence:** 4

**Summary:**

This work studies the Truncated Back-propagation Through Time (TBPTT) algorithm for training recurrent neural networks (RNNs) on time series task. They analyze the theoretical bounds on accuracy and performance loss when optimizing over subsequences compared to the full data sequence. Their bounds reveal that the burn-in period m serves as an important hyper-parameter in guaranteeing better performance. RNN hidden states are 0 initialized and up to time m, the network does not generate any output due to negatively affected internal transients. This paper provides theoretical bounds (on regret/performance loss) showing how the burn-in period m influences performance, and shows empirical evidence across synthetic and real-world time-series/system-identification datasets that tuning the burn-in phase improves results.

**Strengths:**

- This work analyzes TBPTT with zero initialized hidden states from the lens of regret w.r.t. training benchmark and shows that the burn-in period (m) affects the performance guarantees, both training regret and performance regret.
- Experimental evaluations on the synthetic data survey as a good validation of the proposed bounds. It shows that larger values of the burn-in phase (m) allow for longer transient phases, and enables RNN to exploit the exploration to reach closer to the benchmark output.

**Weaknesses:**

- Although the theoretical bounds are valid in the synthetic data setup, the theoretical analysis starts to dwindle down as we reach towards harder real-world setup
- It is unclear how the assumption 1 holds true in realistic settings with more complex choice of RNNs (such as LSTMs, GRUs, etc.)
- Although theoretically the burn-in period helps in simplifying and analyzing the regret, it is unclear if the burn-in period is really the bottleneck in more sophisticated TBPTT (such as non-zero hidden state initialization, hidden state propagation between sequences, normalization techniques, etc.)

**Questions:**

- Other than the contractive dynamics, are there well known dynamics that would satisfy the assumption 1?
- While the impact of the burn-in period is relatively simple to observe in synthetic data setup, but it comes harder as we evaluate on harder and more challenging real-world time series forecasting problem. Does this imply that the analysis breaks down under this setup?
- In the case of real-world time series and sysid problems, the experimental setup uses LSTMs as the RNN choice, is there any clue or intuition as to whether the assumption 1 is valid in this setup ?
- Since typically in TBPTT, the hidden state is carried over from the previous segment, how far off are the empirical numbers compared to this setup?
- Have you evaluated the effect of N and S in the Tab.1 ?
- Although, this paper analyzes the regret w.r.t. TBPTT performance with the optimal hidden state initialization and the correct dynamics, how would the analysis change if we were to compare V* with a V_b which measures the performance of the BPTT algorithm (i.e., to predict the future, one can look at the entire past)?
- Would it be much harder to transfer the theoretical analysis to other state space models?

---

> ### Author Response · Authors · 2025-11-24
> **Response to Reviewer 4peq (Part 1/3)**
>
> We thank the reviewer for reading our manuscript and summarizing the contributions as well as for providing valuable comments and raising interesting questions. We revised the manuscript accordingly and believe that the changes we implemented to address the reviewers' comments and questions significantly improved the overall quality of the paper. We provide the detailed answers below.
>
>
>
> ## **W1: Our theoretical results in  the context of real-world data sets**
>
> Thank you for highlighting the gap between our theoretical analysis and our experiments on real-world data sets. The main issue here is that the benchmark RNN (trained by solving the optimization problem in (11)) cannot be determined due to the size of the problem. Nevertheless, we implemented two main changes in the revised manuscript to narrow the gap between our theory and real-world applications:
>
> 1. We made the theoretical requirements our our results more practical by refining (weaken) Assumption 1 (see W2 below). We numerically tested the TBPTT-trained RNN models after training for complying with Assumption 1, which was always found to be the case.
>
> 2. We significantly extended our experiments by considering various combinations of window length *N* and burn-in phase *m*. Moreover, we made the tuning of *m* more systematic by considering additional validation data sets. In addition, we considered two alternative methods to compare the training results with: stateful TBPTT training (i.e., passing the hidden state between subsequences during training) and full BPTT (see also W3 below). Figure 4, Figure 6 in the Appendix, and Table 1 in the revised version of our manuscript show the newly obtained training results, where we can clearly observe a dependency of *N* and *m* on RNN performance as suggested by our theory.
>
> ## **W2: Assumption 1 for complex RNN architectures**
>
> Thank you for raising this important point. First, we have refined Assumption 1 by weakening it to the core property we actually require for establishing Theorems 1 and 2 - namely, by restricting it to hold only along input sequences of the training data rather than for all possible inputs. Consequently, it is possible to either enforce it during training, or to check it after training.
>
> For the former, note that it is a sufficient condition that the RNN forward dynamics (evaluated on training input sequences) are contractive, and one could apply existing methods from the literature to ensure contractivity during training. In the presence of LSTMs, for example, this boils down to performing simple row-wise normalization of certain weight matrices after each gradient update, which can be done very efficiently, see, e.g., Miller & Hardt (2019) for details. However, although technically possible, the respective conditions on the weight matrices are conservative, which may limit expressiveness of the respective RNN (we actually tested this for certain data sets considered in our experiments, which unfortunately lead to significantly worse performance).
>
> As simple alternative, we numerically tested the RNN models after training for compliance with the property required by Assumption 1 (equation (12) in the revised manuscript), which was always found to be the case (we randomly sampled initial conditions for two different cell states, evaluated the forward dynamics on input (sub-)sequences of the training data, and tested if the RNN output differences satisfy (12) for some pre-defined constants *C* and *λ*). Note that this is not restricted to any RNN architecture.

---

> ### Author Response · Authors · 2025-11-24
> **Response to Reviewer 4peq (Part 2/3)**
>
> ## **W3: Alternative TBPTT methods**
>
> Thank you for highlighting the lack of comparison to more sophisticated TBPTT methods. We fully agree with the reviewer that this is an important point that should be addressed.
>
> To this end, we extended our experiments by additionally performing *stateful* TBPTT training, where the hidden states are passed between subsequences during training instead of being initialized to zero. Indeed, we find that stateful TBPTT training outperforms zero initialization in some cases where long-term dependencies are crucial, as expected. The Solar-Energy data set  in Table 1, for example, nicely illustrates this: especially for a small lookback windows (e.g., N=48), stateful TBPTT training performs much better on the test data as TBPTT training with zero initialization, which indicates that N=48 is simply too short to predict F=96 future values when using zero initialization. This is also validated by the fact that this performance gap decreases when N is increased.
>
> However, we also want to emphasize that TBPTT with zero initialization can still have clear advantages over more sophisticated TBPTT approaches, especially from a numerical point of view. Namely, the randomness of the training algorithm (which is not possible for stateful TBPTT training, where the chronological order of the subsequences must be preserved to pass the correct hidden states) is advantageous, as it can help escape from local minima and generally renders the training process more numerically stable, robust, and efficient, see also the discussion in Remark 1. This effect is also evident during our experiments, where we observe that TBPTT with zero initialization generally converges much faster and more robust compared with stateful training, and ultimately outperforms it in some cases in Table 1 (we compare the results after performing the same number of SGD iterations).
>
> Nevertheless, we believe that extending our theory to stateful training (or other more sophisticated TBPTT methods) is an interesting and important topic for potential future work.
>
>
> ## **Q1: Dynamics satisfying Assumption 1**
>
> First, we want to emphasize that contractive dynamics are sufficient for satisfaction of Assumption 1, but not necessary. Moreover, we have revised Assumption 1, which now requires a weaker, data-dependent property, defined only along input sequences of the training data (in contrast to arbitrary inputs). Consequently, it is possible to test the RNN models after training if they satisfy this property, see W2 above for details. This is not restricted to certain RNN architectures, but generally can be applied to any state-space model.
>
> ## **Q2: Theoretical analysis in the context of real-world time series forecasting problems**
>
> Thank you for this question! We believe that our analysis still provides valuable intuition also for real-world time series forecasting problems. First, we have numerically tested the trained RNN models for compliance with Assumption 1, which is the main prerequisite for our theoretical results. Moreover, in the revised experiments, we now provide the training results for various combinations of window length *N* and burn-in phase *m*, and we can observe a strong dependency of forecasting performance on these parameters. Moreover, we have made the tuning of the burn-in more systematic, by considering an additional pre-training/tuning phase. The training results indicate that it is overall beneficial to tune the value of *m* rather than keeping the standard configuration *m*=*N*-1, as suggested by our theory. Choosing *m*<*N*-1 for time series forecasting can be interpreted as additional regularization, which turns out to be numerically advantageous when performing the gradient updates during training.
>
>
> ## **Q3: Assumption 1 for LSTMs in the experiments**
>
> We have numerically tested the trained LSTMs whether they satisfy the required property, which was always found to be the case (see W2 for further details).
>
> ## **Q4: Comparison with stateful TBPTT training**
>
> Thank you for this question! We now provide training results for this setup in Table 1 in the revised version of our manuscript. As expected, passing the hidden states during training can have clear advantages, especially when long-term dependencies are crucial (see the test results in Table 1 for the Solar-Energy data set). According to intuition, this gap can be reduced by enlarging the TBPTT window *N*, see also W3 above for more details. However, we also observe that in some cases, the numerical advantages of using zero initialization outweigh the benefits of information forwarding, as the overall training algorithm enjoys additional randomness and has a positive effect on generalization (see, for example, the test results on the Traffic data set in Table 1). In the revised version of our manuscript, we have added a discussion in Appendix B.2 in this regard.

---

> ### Author Response · Authors · 2025-11-24
> **Response to Reviewer 4peq (Part 3/3)**
>
> ## **Q5: Experiments for N and S**
>
> Thank you for for this suggestion!  We have revised our experiments accordingly. Specifically, in Table 1 and Figures 4 and 6, we now provide training results for various combinations of *N* and *m*. As expected, the results improve if *N* is increased, which also requires larger training times (we now also report the training times in Figure 4). Interestingly, from Figures 4 and 6 we can observe that the impact of *m* on the RNN performance is qualitatively very similar, regardless of the respective value of *N*.
>
> During the experiments, we generally keep the value of *S* to a fixed constant *S=T-N+1* to leverage as many different subsequences as possible and ensure a fair comparison between the considered methods and parameter choices. We believe that it is not truly meaningful here to explore various options for the number of subsequences used and how they might overlap, as this strongly depends on availability of additional prior knowledge and expertise about the data sets in order to decide which sequences actually contain information and are important for training, and which do not. However, we believe that this is indeed relevant in certain scenarios (especially for very long training sequences), where we expect that a proper choice of the subsequences used for training (and hence the value of *S*) can significantly accelerate the training procedure without sacrificing overall performance. In the revised version of our manuscript, we have added a paragraph in Appendix B.2 in this regard.
>
>
>
> ## **Q6: Benchmark training**
>
> We thank the reviewer for raising this interesting question. In fact, the considered benchmark problem in (11) is actually very close to the setup the reviewer refers to, although the optimization objective in (11a) is formulated in terms of TBPTT performance. However, the additional constraint in (11b) ensures that the corresponding benchmark solution follows one hidden state sequence over the entire input sequence of length *T*, similar as in the BPTT algorithm. As a consequence, the parameters are optimized by considering the entire past. Moreover, we could also reformulate the optimization objective into a form which is very close to the performance of the BPTT algorithm, with the only difference being an additional time-varying weighting factor due to the overlap of the subsequences used in TBPTT.
>
> In the revised version of our manuscript, we have re-formulated the discussion below (11) in order to clarify the benchmark problem.
>
> ## **Q7: Extensions to other state-space models**
>
> Thank you for this question! In fact, the general RNN presented in (1)-(2) covers *any* parameterized nonlinear state space model, and our theoretical analysis can be directly applied to any of those. This therefore also includes, for example, classical system identification problems (Ljung, 1999), in which additional prior (physical) knowledge is usually incorporated in order to learn gray/white box state-space models. In the revised version, we have added a corresponding comment below (3) in this regard.

---

### Official Review · Reviewer_5iFc · 2025-11-04

**Soundness:** 3
**Presentation:** 2
**Contribution:** 3
**Rating:** 8
**Confidence:** 4

**Summary:**

This paper propose a method called burn in phase in training RNN using truncated backpropagation through time (TBPTT) that can improve the performance of existing TBPTT method. The back propagation through time training method for RNN is suffered from the recurrent  unrolling. The TBPTT tries to improve the training speed as a cost of accuracy. It divides long sequences into shorter subsequence and increase the number of mini batch which can be parallelized. However, this method will lower the accuracy as each subsequence has a different initial hidden state compare the original sequence. The author propose a method that can reduce this error. The rationale is that for RNN with decaying memory, it will gradually forget the initial condition, thus by using a loss that discard the first m time indices. The performance can be improved. The author provides theoretical proofs for their proposed method. As well as experimental verifications.

**Strengths:**

The proposed method is easy yet effective. The author provides theoretical proofs for their proposed method.

Theorem 1.1 An error bound is developed between the model with different initial hidden states, shows that the difference converges if the model has decaying memory (lambda  small)
Theorem 1.2 A bound for training regret is developed. This is the difference between ideal case approximation error where an ideal initial state is known and the approximation of using subsequence with burn in.


Theorem 2.1 Shows 1 on full sequence
Theorem 2.2 Shows 2 on full sequence

**Weaknesses:**

Some of the concept and terms does not have enough motivation, and the intuition and explanation is not very clear for the Theorems, which can cause non expert hard to follow the results.

There is no recall of notations in Theorem part. Need frequently refer back to find notations.

Certain part of the experiments can be improved.

Details please see questions.

**Questions:**

1. Line 307-308 says that "Comparing the corresponding solutions therefore allows us to investigate and quantify the effects of using zero initialization in TBPTT compared to using the optimal initializations."

Why is the optimal initialization as defined in equation (11) is minimized over {h_t}.  $h_0$ is the initial condition, which is a free variable that can be optimized. But how does $h_1$ to $h_T$ be optimized? does that depend on the inputs and $\theta$. I do not understand the state in line 305-306 which says that the additional constraint forces the underlying hidden state sequences to be a subsequence of the same hidden state sequence.


It appears to me that the most optimal setting is each subsequence $i \in \{1,...S\}$ has its own initialization,so the minimization is taken over $S$ number of initial hidden state for the $S$ different subsequences.



2. It appears to me that Assumption 1 is saying RNN has exponential memory decay. This means that the dependence on initial condition decays exponentially which is characterized by $\lambda$. Is this equivalent to a more simple assumption which assume the largest eigenvalue of $W_{hh}$ is smaller than 1. And does the $\lambda$ has any relation with the largest eigenvalue of $W_{hh}$?


3. The notation in Theorem 1 means that $\theta^\*$ and $\theta^b$ are any parameters from the parameter space. But the superscript $*$ and $b$ are used before to indicate model using zero initial and optimal initial. The notation here need to be clarified.

4. Is "batch-averaged output differences" a standard term? If not, please explain the physical neaning of this quantity.

5. "As this constant does not depend on N , they must become smaller if N is increased" What is "they" refers here, this argument need to be more
rigorous

6. The RHS of Theorem 1.2 has a term $\frac{\lambda^m}{N-m}$, this function first decays and then increases on interval $(0,N)$. And the best possible $m$ depends on $\lambda$ and can be explicitly solved. Why line 356 to line 358 only discuss this optimal $m$ vaguely? Is it true that the this optimal $m$ results from the upper bound is not the actually optimal choice of $m$?

7. What is the trade-off discussed in line 380 to line 384. Currently what I can infer from the discussion here is that we want larger $m$, which requires larger $o_{min}$, which results in larger $S$.  This means that larger $m$ results in larger $S$.
    Firstly, my question is do we always want a larger $m$.
    Secondly, is the argument here rigorous. Actually when $m$ is large, the upper bound exploded, however, this does not necessarily means that LHS also explode. To be rigorous, I think a lower bound is needed. If not, at least there should be some numerical evidence.


8. Regarding the synthetic experiment:

    a. For the synthetic data experiment. How does it verifies your theories. Is the exponential decay rate follows that $\lambda$ of the unknown system.
    b. Is it even possible to know the $\lambda$ of the unknown system.
    c. You talked about best choice of $m$ depends on $\lambda$, but from this experiment, can only see larger $m$ the better. What if when $m$ is very large? Does it make the model behave bad as predicated by the upper bound, if not, this means that the bound is too loose.
    d. Is there any experiment to show the trade-off mention in line 380 to 384.

    e. Why do you conduct experiments on a unknown system, which cannot fully verify your results. Is it more reasonable to conduct experiments on a known system which you can control the $\lambda$, to fully support your theoretical results.

---

> ### Author Response · Authors · 2025-11-24
> **Response to Reviewer 5iFc (Part 1/3)**
>
> We thank the reviewer for the careful review of our paper, summarizing the main contributions, and providing many valuable comments and questions that helped us improve the clarity and presentation of the paper. Below we provide detailed answers to all questions and comments.
>
>
> ## **W1 and W2: Overall presentation of our theoretical results**
> Thank you for pointing out the lack of clarity in the presentation of our results. In this regard, we have revised the corresponding parts of Section 4.1 to better motivate the benchmark problem in (11). Moreover, we have revised the theorem statements as well as their discussions and implications.
>
> ## **W3: Experiments**
>
> We have significantly revised Section 5 and improved the experiments. First, we clarified the role of our main assumption (Assumption 1) in the synthetic experiment and how we actually verified/enforced it during training. Moreover, we have extended Section 5.2, where we consider RNN training on public data sets (see our general reply for further details). Here, we observe a strong influence of the burn-in phase *m* on the training process, and that treating *m* as an additional hyperparameter is beneficial for overall RNN performance.
>
> ## **Q1: Regarding the benchmark problem (11)**
>
> Thank you for raising these questions. First, we want to recall that we measure RNN performance in Section 4 using the performance index P as defined in (9), that is, the MSE of the generated network outputs after processing the whole training input sequence of length *T*. The TBPTT optimization objective in (10), in contrast, aims at minimizing the averaged MSE of S output sequences of length *N*, generated using subsequences of the input training data.
>
> The benchmark problem in (11) is designed to leverage both: the original TBPTT optimization objective from (10), and the knowledge that the outputs should be ultimately generated from one single hidden state trajectory. Consequently, free optimization variables of the problem in (11) are the network parameters *θ* and the initial hidden state *h*\_0, as they uniquely determine the full hidden state sequence {*h*\_1,...,*h*\_T} using the forward dynamics of the network. The problem in (11) actually represents an equivalent multiple shooting version of this configuration, where all hidden states are treated as optimization variables, but which are subject to the constraint in (11b), namely the forward dynamics of the network.
>
> We have revised the corresponding paragraph in the revised version of our manuscript to emphasize this more clearly. To this end, we also labeled the optimization objective and the optimization constraint of the benchmark problem in (11) individually using the respective subequations (11a) and (11b).
>
> **Individual hidden states:** The reviewer raises an interesting point in the latter part of Question 1, suggesting that each hidden state sequence appearing in (11) should rather have its own initialization. We fully agree with this assessment - if one would solely be interested in achieving the lowest value of the optimization objective in (10)/(11) at any cost. However, we want to clearly emphasize that this does *not* ensure good performance in terms of the criterion in (9), i.e., when processing the full input sequence of length *T* - indeed, if *T* is much larger than *N*, this may even significantly degrade overall performance, making it unsuitable as a benchmark. Nevertheless, we appreciate this comment, as it corresponds precisely to the problem formulation that is at the core of our theoretical analysis in Appendix A. We would like to encourage the reviewer to take a closer look at the problem formulation in (16) and (20).

---

> > ### Author Response · Authors · 2025-11-24
> > **Response to Reviewer 5iFc (Part 2/3)**
> >
> > ## **Q2: Assumption 1**
> >
> > We thank the reviewer for this question. We agree that Assumption 1 essentially requires exponential forgetting of network initialization, at least if we limit this to the parts that actually reach the output during training (Assumption 1 in the revised version is actually much weaker than requiring contractive forward dynamics). Hence, it is *not* equivalent to assuming that the weights *W*\_hh have eigenvalues smaller than 1 (also because we do not restrict our analysis to certain architectures such as the exemplary RNN in (3)). However, the reviewer raises an interesting point; indeed, for certain RNN architectures such as in (3), enforcing that *W*\_hh has eigenvalues smaller than 1 actually is sufficient for Assumption 1 to hold true, as this essentially restricts the model to have contractive forward dynamics. In this case, the value of *λ* can also be directly related to the largest eigenvalue of *W*\_hh, compare Section 5.1 and see also work by Miller & Hardt (2019) for more details on how to ensure model stability/contractivity of RNNs during training.
> >
> > In the revised version of our manuscript, we have revised the discussion below Assumption 1 in this regard, now clearly highlighting the relation to the concept of contractivity and how Assumption 1 can be checked in practice.
> >
> > ## **Q3: Theorem statements**
> >
> > Thank you for pointing out this lack of clarity. Indeed, with _θ_^* and *θ*^b, we are referring to the solutions of the TBPTT problem in (10) and the benchmark problem in (11), rather than considering any two parameters from the parameter space. In the revised version of our paper, we reformulated the statements of Theorems 1 and 2 to avoid potential  ambiguities.
> >
> >
> >
> > ## **Q4/Q5: Batch-averaged output differences**
> >
> > We thank the reviewer for highlighting weaknesses in our wording. With "batch-averaged output differences", we simply wanted to refer to the left hand side of the property 1 in Theorem 1, that is, the averaged squared errors between the predictions generated by the TBPTT solution _θ_^* and the benchmark solution *θ*^b over all mini-batches. However, we agree with the reviewer that this term is not standard and hence may cause confusion for the reader. This also applies to the ambiguous formulation pointed out in Question 5; with "they", we also wanted to refer to the sum of the output differences. In the revised version of our manuscript, we have revised the corresponding discussion below Theorem 1, improving readability and clarity.
> >
> >
> > ## **Q6: Optimal value of _m_**
> >
> > We thank the reviewer for raising this important question. First, we agree that If an upper bound for *λ* is known/enforced during training, the optimal value of *m* minimizing the regret estimates from Theorems 1 and 2 could, in principle, be computed explicitly. However, note that the estimates correspond to worst-case upper bounds on the regret and hence are generally conservative, as the quantitative property given by Assumption 1 is conservative by nature (it must be designed to cover the slowest possible decay ever encountered during training). Our theoretical results should therefore be understood as qualitative tuning guidelines rather than quantitative ones. We have added a paragraph below Theorem 2 in this regard.
> >
> >
> > ## **Q7: Trade-off between large *S* and large *m***
> >
> > Thank you for raising these points. First, we want to make clear that the choice of *m* is completely application dependent, and there is no general answer whether one would always prefer larger or smaller values of *m* (compare also the revised experiments in Section 5); instead, our main conclusion is that *m* should be treated as hyperparameter rather than fixing it a priori. Moreover, note that the bounds provided by Theorems 1 and 2 generally cannot explode, as all parameters *S*, *m*, etc. are bounded due to the fact that the training data set is finite; for example, we necessarily have that *m*≤*N*-1, *S*≤*T*-*N*+1, *N*≤T.
> >
> > While we agree that the overall trade-off between performance, *S*, and *m* is interesting from a theoretical perspective, we would like to use the limited space below Theorem 2 to discuss more relevant points, namely (i) the interplay between *λ* and *m*, and (ii) the qualitative nature of our theoretical results, as part of the newly added paragraph in response to Question 6 of this reviewer.

---

> > > ### Author Response · Authors · 2025-11-24
> > > **Response to Reviewer 5iFc (Part 3/3)**
> > >
> > > ## **Q8: Synthetic experiment**
> > >
> > > We thank the reviewer for being interested in our synthetic experiment. Below we provide the answers to the detailed questions (a)-(e).
> > >
> > > - **(a,b,e)** First, we want to emphasize that Assumption 1 (where the decay rate _λ_ appears) is not a property of some unknown underlying data-generating system - it is rather a property of the trained RNN model. Consequently, we do *not* require the existence of such system, nor do we require any conditions in this regard. We have revised the discussion below Assumption 1 to emphasize this more clearly.
> > >
> > >   In the synthetic experiment, we can indeed control the value of *λ*, and we do so by enforcing ||*W*\_hh||<0.999 during optimization, compare also our answer to Question 1 of this reviewer. Consequently, Assumption 1 is satisfied, and our theoretical results hold true.
> > >
> > > - **(c)**  We agree that from the experiments, we find that the maximum value of *m* yields the best results (i.e., *m*=*N*-1), see the right plot in Figure 3, where we plot the overall performance in terms of the corresponding measure *P* as defined in (9). This is actually in line with our theory, as choosing larger values of *m* reduces the developed *performance regret* estimates provided by Theorem 2. Here, this is also intuitive, as larger values of the burn-in phase *m* allow for longer transient phases, and ultimately enables the RNN to exploit the exploration to reach closer to the benchmark output, compare also Figure 5 in Appendix B.1.
> > > - **(d)** In all our experiments, we chose the maximum number of subsequences, i.e., *S*=*T*-*N*+1. This is also practically relevant, as this uses as many different subsequences as possible during training. Here, we want to emphasize that choosing smaller values of *S* generally requires availability of additional prior knowledge and expertise about the respective data sets in order to decide which subsequences actually contain information and are important for training, and which do not. We believe that this is indeed relevant in certain scenarios (especially for very long training sequences), where we expect that a proper choice of the subsequences used for training (and hence the value of *S*) can significantly accelerate the training procedure without sacrificing overall performance. However, a thorough comparison for different (meaningful) choices of *S* is hence out of the scope of this paper; but we generally agree with the reviewer that investigating the potential impact on overall performance seems important, and could be an interesting subject for future work. In the revised version of our manuscript, we have added a paragraph in Appendix B.2 in this regard.

---

> > > > ### Comment · Reviewer_5iFc · 2025-11-26
> > > >
> > > > I appreciate the author for the reply and the updated manuscript is much clearer. Thanks for addressing my main concerns, I will maintain my score.

---

> > > > > ### Author Response · Authors · 2025-11-26
> > > > >
> > > > > Thank you very much for your positive feedback! Many thanks again for your detailed review, which was very helpful in improving the overall quality of the work.

---

### Author Response · Authors · 2025-11-24
**General reply and major changes**

We thank all reviewers for their careful review of our paper and for their useful suggestions and constructive criticisms. We have considered each comment and revised the manuscript accordingly. For convenience of the reviewers, all the significant changes in the revised manuscript are highlighted in red. The major changes can be summarized as follows.

**1. Experiments:** We have significantly extended our experiments performed in Section 5.2. In particular:
   - For all data sets, we have trained the RNNs using various combinations of the truncation window length *N* and burn-in phase length *m*.
   - We have tuned the burn-in phase length *m* based on additional validation data sets.
   - We have numerically verified Assumption 1 (output stability of the trained network model) for the trained RNNs along input sequences of the training data.
   - For comparison purposes, we have additionally performed BPTT and stateful TBPTT training.
   - We have reported and compared the respective training times.

   In all experiments, we observe a strong influence of the burn-in phase on the training process, and proper tuning can lead to a reduction of the MSE achieved on the training and test data of more than 60% in some cases.

**2. Presentation of our theoretical results:** We have revised many discussions and the overall presentation of our theoretical results. Moreover, we have refined our main theoretical assumption, namely Assumption 1 - now requiring the corresponding output stability property only *along sequences of the training data*, rather than uniformly for all possible inputs. Consequently, this property can be enforced during training, or numerically checked after training (as we did in our revised experiments).

---

> ### Author Response · Authors · 2025-12-04
>
> We have uploaded a revised version of our code for the experiments performed in Section 5.2 to ensure reproducibility. This includes the source code for RNN training as well as the training results presented in the revised version of the manuscript, including the trained models and post-processing scripts. Moreover, we have updated the manuscript after repeating the experiments, which was necessary due to several minor bug fixes in the code (without significant impact on the overall findings).

---

### Meta-Review · Area_Chair_rQQU · 2026-01-07

**Summary:**

The paper studies truncated backpropagation through time (TBPTT) for training recurrent neural networks and focuses on the role of the burn-in phase, namely the initial part of each truncated subsequence whose outputs are excluded from the loss. The core contribution is a theoretical analysis that quantifies the performance and regret gap between TBPTT (with zero-initialized hidden states) and an idealized benchmark that effectively corresponds to full-sequence training. The analysis shows that, under an output stability / exponential forgetting assumption, the choice of burn-in length $m$ critically affects the regret bounds, and that increasing or properly tuning $m$ can significantly reduce the performance gap.

Beyond theory, the paper provides empirical validation on synthetic system identification tasks and several real-world time-series benchmarks. These experiments demonstrate that the burn-in phase has a large practical impact and that tuning it as a hyperparameter rather than fixing it heuristically can substantially improve both training and test performance, in some cases dramatically.

**Reviewer Concerns:**

Initial assessments from reviewers were mixed, with one strongly positive score. The concerns raised by reviewers were as follows:

- Clarity (raised by 5iFc): One reviewer raised extensive issues about unclear notation, insufficient intuition behind theorems, ambiguous definitions (e.g. optimal initialization, benchmark problems), and informal arguments in the theoretical discussion. Several bounds and trade-offs were presented without clear explanation of what quantities mean or why conclusions follow. The authors made substantial expository revisions, rewriting theorem statements, clarifying the benchmark optimization problem, explicitly distinguishing TBPTT vs benchmark solutions, and cleaning up ambiguous wording. Reviewer 5iFC explicitly acknowledged that the changes improved clarity and maintained their positive score.

- Assumption 1 (raised by 4peq, 5iFc): Reviewers questioned whether the key assumption is realistic, especially for modern architectures, and whether it meaningfully holds beyond synthetic settings. There was scepticism that the theory meaningfully applies when assumptions are unverifiable or only weakly connected to practice. The authors weakened the assumption to hold only along training data, argued that it can be enforced via contractivity constraints or checked post hoc, and reported numerical verification for trained models, including LSTMs. The assumption is now significantly more defensible.

- Theoretical bounds (raised by 4peq, xtPM): Reviewers questioned whether the regret bounds actually explain or predict performance in challenging real-world datasets, noting that theoretical insights appeared strongest in synthetic experiments and less informative as tasks became more complex. The authors acknowledge that the bounds are qualitative in nature, and not quantitatively tight, repositioning them as tuning guidelines. They expand experiments across datasets, window lengths, and burn-in values, showing consistent qualitative trends. This is pretty typical, so I’m happy to say the issue seems resolved.

- Stronger baselines (raised by 4peq, xtPM): Several reviewers criticized the lack of comparison to full BPTT and more sophisticated TBPTT variants (e.g. stateful hidden-state propagation), questioning whether burn-in is truly a bottleneck relative to these alternatives. The authors added comparisons to full BPTT and stateful TBPTT, reported training times, and discussed trade-offs. Results show that tuned burn-in narrows the performance gap to BPTT and that zero-initialized TBPTT can sometimes outperform stateful variants due to optimization stability. I consider these concerns addressed.

- Tuning (raised by xtPM): Reviewer xtPM questioned whether systematic tuning of $m$ undermines the computational advantage, and whether hyperparameters tuned on training data generalize to test performance. The authors added validation-based tuning, training-time comparisons, and test results. They argue that TBPTT is often necessary regardless (BPTT infeasible) and that tuning costs are acceptable in practice. Some better tuning approaches would have been ideal, but otherwise I believe this is fine.

**Reviewer Scores:**

There are only three reviewers for this article, so the presence of a high score of 8 weighs heavily on the final decision. The other two reviews currently stand at a 4, and the corresponding reviewers did not get the opportunity to respond. I feel that all of the concerns raised have been adequately addressed, so it seems plausible that the scores could have increased further. After substantial revisions by the authors, I believe that the current revision is acceptable for publication.

---

### Decision · Program_Chairs · 2026-01-26

Accept (Poster)